# The Effects of Phosphorylation and Microwave Treatment on the Functional Characteristics of Freeze-Dried Egg White Powder

**DOI:** 10.3390/foods11172711

**Published:** 2022-09-05

**Authors:** Zahra Jafari, Mohammad Goli, Majid Toghyani

**Affiliations:** 1Department of Food Science and Technology, Isfahan (Khorasgan) Branch, Islamic Azad University, Isfahan 81551-39998, Iran; 2Laser and Biophotonics in Biotechnologies Research Center, Isfahan (Khorasgan) Branch, Islamic Azad University, Isfahan 81551-39998, Iran; 3Department of Animal Science, Isfahan (Khorasgan) Branch, Islamic Azad University, Isfahan 81551-39998, Iran

**Keywords:** phosphorylation-microwave treatment, egg white protein powder, physicochemical characterization, particle-size distribution, scanning electron microscopy, Fourier transform infrared spectroscopy

## Abstract

The effects of phosphorylation pre-treatments at 1.5, 2.5, and 3.5% levels, as well as microwave application at 200, 400, and 700 watts levels for 2 min, on the functional parameters of egg white powder obtained by the freeze dryer procedure were investigated. P1.5-M200 had the highest oil-holding capacity, emulsion stability, and emulsion activity, while P2.5-M200 had the highest foam capacity. The P2.5-M400 had the largest particle size, and P3.5-M200 had the highest degree of phosphorylation and protein solubility. On the other hand, P3.5-M200 had the highest solution viscosity by 1% (*w*/*v*), water-holding capacity, and foam stability, in the treatments that used phosphorylation and microwave treatment simultaneously. FTIR spectroscopy of the unfolding structure of egg white protein revealed changes in the protein’s secondary structure, such as the development of β-sheets and β-turns, as well as the binding of negatively charged phosphate groups on the serine, threonine, and tyrosine side chains. The phosphorylation and microwave treatments reduced the particle size of the egg white protein powder while increasing the surface area of the protein molecules, according to SEM analyses.

## 1. Introduction

Egg white contains a variety of proteins, including ovalbumin, ovotransferrin, ovomucoid, ovomucin, and lysozyme [1]. Eggs are highly nutritious presenting many functional properties for human consumption [2]; however, egg white is the most important component for the food industry [3]. Egg white powder has become a popular commercial product as an alternative for fresh and liquid eggs because it can avoid microbiological and oxidative deterioration and has cheaper packaging costs [3]. Proteins’ functional properties can be altered by physical, chemical, and enzymatic treatments. Since chemicals can be hazardous to human health, their usage to modify proteins in the food industry is strictly prohibited [4]. The use of enzyme modification is often restricted due to the high cost of enzymes as well as difficult and complex operating conditions. As a result, physical protein modification (thermal processing, i.e., microwave treatment) is of considerable interest [5]. The current challenge is to identify novel processing techniques (such as ultrasound, pulsed electric field, high pressure processing, radio frequency, ultraviolet light, microwave, and cold plasma for egg products) that improve not only the intrinsic functional properties of eggs or their components, but also the product’s quality. These innovative methods are recognized for their advantages over thermal treatments, particularly in protecting the heat-sensitive nature of eggs and egg products. The availability of alternative processing methods has substantially increased the structural features, technological functionality, nutritional value, and safety of eggs and egg products [6].

Protein phosphorylation is a typical post-translational protein modification occurring in nature. In general, amino acid residues are phosphorylated by the addition of a covalent phosphate group bound by the protein kinase by serine, threonine, and tyrosine. Phosphorylation modifies the amino acid side chain by adding a charge and a hydrophilic group. According to various studies, phosphorylated proteins have higher solubility than non-phosphorylated proteins because they have a larger negative charge [7,8]. The phosphorylated protein molecules have a strong electrical repulsion force, which increases their solubility in water. After egg white phosphorylation, powder particle size decreased but wettability, dispersibility, and solubility rose [3]. The most prevalent phosphorylation techniques are dry heating or bath water heating [9]. Microwave-assisted phosphorylation significantly accelerated the procedure, and phosphorylation of egg white protein resulted in improved functional properties. The effect of microwave-assisted phosphorylation on powder characteristics, on the other hand, is unclear [3].

For a few years, microwave-assisted chemical procedures have been employed to achieve chemical reactions. When compared to traditional heating techniques, chemical synthesis with microwaves help can significantly shorten reaction duration while also improving yield and product purity. Microwave-assisted approaches have previously been used by certain researchers in the chemical alteration of dietary protein [10]. When compared to the traditional dry-heating approach, microwave treatment can significantly shorten reaction times and speed up the phosphorylation process. Li et al. [3] used a spray dryer to produce three different egg white protein powders (untreated, microwave-alone treated, and microwave-assisted phosphorylation modification by sodium tri-phosphate (STP or Na_3_PO_4_)) and investigated their physicochemistry and rehydration behavior.

There are several drying techniques available for producing sodium tri-phosphate egg white powder (STP-EWP), and these drying methods should be chosen based on drying efficiency and dried product quality. It has been found that freeze-dried proteins retain their natural structural shape better because they are subjected to fewer heat and water evaporation-related stresses. The freeze-dried particles are more porous and shrink less. According to Lili et al. [11] freeze-dried STP-EWP powders outperformed spray-dried STP-EWP powders in terms of solubility, emulsion stability, water-holding capacity, oil- and water-absorption capacity, and heat gel strength. Freeze drying was found to be the most effective way for producing modified EWP powders with enhanced functional characteristics.

For the first time, the combined effects of phosphorylation with Na_2_HPO_4_ (1.5, 2.5, and 3.5%), microwave treatment with power (200, 400, and 700 watt), and finally freeze drying (0.04–0.06 mbar & −50 °C) on the functional characteristics including protein solubility, foam capacity, foam stability, emulsion activity, emulsion stability, water-holding capacity, oil holding capacity, viscosity of solution 1%, as well as degree of phosphorylation, particle-size distribution, FTIR phosphorylation analysis, and SEM morphological analysis of egg white protein powder were investigated.

## 2. Materials and Methods

### 2.1. Materials

A total of 360 fresh chicken eggs were provided from a local farm (Simorgh LTD, Isfahan, Iran). The analytical-grade ingredients were disodium hydrogen phosphate (DSP, Na_2_HPO_4_), citric acid dihydrate, nitric acid, sulfuric acid, perchloric acid, trichloroacetic acid, potassium bromide, bovine serum albumin, and the rest of the reagents (Merck, Darmstadt, Germany).

### 2.2. Preparation of Egg White Protein Solution

The white of the fresh eggs was carefully separated from the yolk. The egg whites were stirred for 1 h at 4 °C using a magnetic stirrer to create homogeneous egg whites. The egg white was then tripled in volume with distilled water and agitated for 1 h at 4 °C using a magnetic stirrer. This solution was used to make microwave-assisted phosphorylated egg white protein powder (PM-EWP) treatments [3].

### 2.3. Preparation of Microwave-Assisted Phosphorylated Egg White Powder

DSP was dissolved in the egg white protein solution to generate 15, 25, and 35 g L^−1^ mixed solutions (i.e., 1.5, 2.5, and 3.5% phosphorylated treatments) at pH 8 by lowering the pH with citric acid dihydrate (0.033 M, pH 2.2). Then the solutions were microwaved for 2 min with powers of 200, 400 and 700 watts. The solutions were immediately suspended in cold water at 0 °C for 4–5 min in order to prevent excessive reactivity and bring the temperature of the solution up to 10–15 °C. The extra phosphates were then precipitated as calcium phosphate by adding 19, 32, and 44 g L^−1^ of calcium chloride to 1.5, 2.5, and 3.5% phosphorylated solutions, respectively. To remove excess calcium phosphate, the solutions were centrifuged at 3000× *g* for 5 min at room temperature. At a temperature of −50 °C and an air pressure of 0.04–0.06 mbar, the supernatant was dried in a freeze drier [3]. The control treatment, also known as P0-M0, refers for no phosphorylation and microwave treatment. Other treatments, such as P3.5-M700, indicate for 3.5% phosphorylation and 700 watts of microwave.

### 2.4. PM-EWP Functional Properties Determination

#### 2.4.1. Viscosity

A Brookfield rotating viscometer with a cylindrical spindle (Brookfield Engineering Labs Inc., Middleboro, MA, USA) (LV-61) was used to evaluate the viscosity of PM-EWP samples (1%). The spindle revolved at 60 rpm. An aliquot of 50 mL was added to the 80 mL flask before testing and allowed to equilibrate at 25 °C [12].

#### 2.4.2. PM-EWP Solubility

With minimal adjustments, the technique of Salvador et al. [13] was used to determine PM-EWP solubility. Bovine serum albumin was utilized as a reference protein in this calibration, and protein content was determined using a biuret test at a wavelength of 572 nm. After being freeze-dried (1 g), the egg white powders were diluted in 100 mL of distilled water and centrifuged at 5000× *g* for 30 min at 4 °C. Protein solubility was calculated using grams of soluble protein per 100 g of egg white protein powder.

#### 2.4.3. Foaming Properties

A 50-mL aliquot of PM-EWP solution (10 mg mL^−1^ solution) was put in a graduated glass cylinder (internal diameter 72 mm) and whipped for 4 min with a laboratory homogenizer at 9500 rpm (Yellowline, DI 25 basic, Ica Works Inc., Wilmington, MA, USA, 600 W, 50 V, 8000–24,000 rpm). The propeller was removed immediately after whipping and the glass cylinder was sealed with parafilm to minimize foam disruption. The foam capacity (FC) was computed by taking the foam volume after whipping (mL) at 0 min and calculating it to a percent of the original foam volume (mL). Stefanovi et al. [14] determined the foam stability as a FS (%) by estimating foam volume after 30 min of standing (mL).
(1)$$\mathrm{Foam}\;\mathrm{capacity}\;\left(\mathrm{FC}\right)\;\left(\%\right)=\frac{\mathrm{Volume}\;\mathrm{after}\;\mathrm{whipping}\;-\;\mathrm{Volume}\;\mathrm{before}\;\mathrm{whipping}\;\left(\mathrm{mL}\right)}{\mathrm{Volume}\;\mathrm{before}\;\mathrm{whipping}\;\left(\mathrm{mL}\right)}\times 100$$
(2)$$\mathrm{Foam}\;\mathrm{stability}\;\left(\mathrm{FS}\right)\;\left(\%\right)=\frac{\mathrm{Volume}\;\mathrm{after}\;\mathrm{standing}\;-\mathrm{Volume}\;\mathrm{before}\;\mathrm{whipping}\;\left(\mathrm{mL}\right)}{\mathrm{Volume}\;\mathrm{before}\;\mathrm{whipping}\;\left(\mathrm{mL}\right)}\times 100\;$$

#### 2.4.4. Emulsifying Properties

PM-EWP dispersion (1% *w*/*v*) was made with sunflower oil in a 1:1 molar ratio and centrifuged for 5 min at 1100× *g*. Emulsion activity was calculated as the ratio of the height of the emulsified layer to the total content in the tube using Equation (3). Equation (4) was used to calculate the emulsion’s stability after heating it at 80 °C for 30 min and centrifuging it at 1100× *g* for 5 min [15].
(3)$$\mathrm{Emulsion}\;\mathrm{activity}\;\left(\%\right)=\frac{\mathrm{Height}\;\mathrm{of}\;\mathrm{emulsified}\;\mathrm{layer}\;\mathrm{in}\;\mathrm{the}\;\mathrm{tube}}{\mathrm{Height}\;\mathrm{of}\;\mathrm{the}\;\mathrm{total}\;\mathrm{content}\;\mathrm{in}\;\mathrm{the}\;\mathrm{tube}}\times 100$$
(4)$$\mathrm{Emulsion}\;\mathrm{stability}\;\left(\%\right)=\frac{\mathrm{Height}\;\mathrm{of}\;\mathrm{emulsified}\;\mathrm{layer}\;\mathrm{after}\;\mathrm{heating}}{\mathrm{Height}\;\mathrm{of}\;\mathrm{emulsified}\;\mathrm{layer}\;\mathrm{before}\;\mathrm{heating}}\times 100$$

#### 2.4.5. Water-Holding Capacity (WHC) and Oil-Holding Capacity (OHC)

First, 1 g of PM-EWP was weighed and stirred for one minute in 10 mL of distilled water or maize oil to determine these properties (Mazola, CPI International, Santa Rosa, CA, USA). The supernatant volume was calculated after centrifuging the protein suspensions at 2200× *g* for 30 min. The g of water retained per gram of material was used to compute the water-holding capacity. In grams of oil retained per gram of protein sample, the oil-holding capacity was reported [16].

#### 2.4.6. Phosphorylation Degree

The phosphorus content of the digest was utilized to compute the total phosphorus content of the protein, which was digested in nitric acid, sulfuric acid, and perchloric acid (10:2:1). To measure inorganic phosphorus (Pi), 5 mL of 10 Mg/mL^-1^ sample solution was mixed with the same volume of 15% trichloroacetic acid and centrifuged at 3000× *g* for 20 min. The quantity of phosphorus bound to proteins was calculated using the difference between total phosphorus and Pi content [17].

### 2.5. PM-EWP Physical Properties Determination

#### 2.5.1. Particle-Size Distribution (PSD)

The PSD of the PM-EWP_s_ was measured using laser light scattering on a Malvern Mastersizer 2000 equipped with a Scirocco 2000 dry dispersion unit (Malvern Instruments, Worcestershire, UK). Moderate particles were added to dry the dispersion device to achieve the required opacity, and the PSD was determined at least three times. Percentile values are represented by the letters d(10), d(50), and d(90). These are statistical characteristics that may be derived directly from the particle size distribution’s cumulative distribution. They represent the particle size below which 10%, 50%, or 90% of all particles are detected. D(4.3) is the volume-weighted mean size. It is the mean diameter, which may be calculated directly from particle size measurements when the recorded signal is proportional to the particle volume. Surface-weighted mean size, i.e., d(3.2) is defined as the diameter of a sphere that has the same volume/surface area ratio as a particle of interest [18,19].

#### 2.5.2. Molecular-Structure Assessment by Fourier Transform Infrared Spectroscopy (FTIR)

At 25 °C, each sample was evaluated using an FTIR spectrometer in the wave number range of 500–4000 cm^−1^ (Nicolet Nexus 470). The instrument’s resolution was 4 cm^−1^, and each sample was scanned using air as the backdrop. In total, 32 scans were used to create each measurement. The materials were mixed with potassium bromide and formed into pellets before being measured [3].

#### 2.5.3. Scanning Electron Microscopy (SEM)

The morphologies of the EWP were imaged using a Tungsten-filament SEM (JSM-6390LV, JEOL, Tokyo, Japan). Prior to SEM, the samples were coated with around 20 nm gold–palladium using a gold sputter module in an argon-filled high-vacuum evaporator. The accelerating voltage was set to 15 kV, and the sputtering period was set to roughly 30 s [20,21].

### 2.6. Statistical Analysis

This study utilized a completely randomized design with two variables. The first variable was the disodium hydrogen phosphate proportion at 1.5, 2.5, and 3.5%, while the second was the microwave device power at 200, 400, and 700 W for 2 min. Three control samples were used: one with no phosphorus and no microwave treatment, one without phosphorus and 700 W microwaves, and one with 3.5% phosphorylation but no microwave treatment. As a consequence, 12 different treatments were investigated. Each test was replicated three times. Physicochemical studies were performed on the first day of PM-EWP manufacturing. Duncan’s multiple range test revealed significant differences at the 95% confidence level (*p* < 0.05) using statistical SPSS software version 16 (SPSS, Inc., Chicago, IL, USA).

## 3. Results and Discussion

### 3.1. Functional Properties

#### 3.1.1. PM-EWP Protein Solubility

Figure 1 shows that protein solubility decreased substantially as microwave power increased from 200 to 700 watts at each level of phosphorylation (%) but significantly increased at each level of microwave power when phosphorylation level increased from 1.5 to 3.5%. P3.5-M0 was the most soluble for the protein. The protein solubility of phosphorylated and microwaved EWP was significantly increased (*p* < 0.05) as compared to the control (untreated) sample, i.e., P0-M0. Protein solubility is the most significant functional characteristic of a protein due to its influence on emulsifying, foaming, and gelling qualities [22]. This phenomenon may be explained by the fact that microwave radiation significantly affected a protein’s solubility when the quantity of phosphate groups in the protein was low. Increased microwave power may cause the hydrophobic group of the protein to be exposed, resulting in a loss in solubility [17,23]. Too-high temperatures caused by increased microwave power may result in protein thermal denaturation, resulting in reduced solubility [3]. Since phosphorylated protein molecules have a higher electrostatic repulsion, they may dissolve more quickly in the solvent. Additionally, the surface charge of the protein molecule increased the interaction of protein molecules with water molecules, resulting in the creation of a hydration layer and soluble aggregates [9].

#### 3.1.2. PM-EWP Foaming Capacity and Stability

Figure 2 shows that increasing the microwave power from 200 to 700 watts decreased the foam production capacity significantly at each level of phosphorylation, and the treatment containing 2.5% phosphorylation had the highest foam production capacity at each level of microwave power. Treatments P2.5-M200 and P0-M700 had the highest and lowest foam production capacity, respectively. The dispersion of gas bubbles inside a continuous liquid or semi-solid phase results in foam in food [24]. Protein solubility is the important factor in protein foaming [23]. Partially opening the protein structure during the microwave process, exposing the hydrophobic and sulfhydryl groups, or partially denaturing the protein, reduced particle size and allowed for better protein absorption at the air-water interface, increasing the protein’s foaming properties [23,25]. Proteins with a high surface hydrophobicity were identified to adsorb extensively at the air–water interface, resulting in a considerable reduction in interface or surface tension, which was used for foaming [17]. The electrical conductivity of microwave-vacuum drying EWP samples may have also been increased; as a result, the distribution of charged, polar, and non-polar residues of the protein molecules was altered, reducing the foaming capacity of the microwave-vacuum drying protein compared to freeze drying samples [26,27].

Increasing the microwave power from 400 to 700 watts greatly enhanced the stability of the foam at each level of phosphorylation %, although the difference between different levels of microwave power of 200 and 400 watts was not significant. At all the same levels of applied microwave power, 3.5% and 2.5% phosphorylation showed the highest and lowest foam stability, respectively. P0-M700 and P3.5-M0 treatments had the highest and lowest foam stability, respectively. According to the statistics, the group with the greatest foaming capacity did not have the best foaming stability. Excessive microwave power might be to responsible for the findings. According to Duan et al. [28], extensive oxidation of egg white protein can result in an increase in foaming capacity but a decrease in foaming stability. The reduction in foaming stability could be due to a drop in the substituents stabilized and more cohesive interfacial coatings formed in the presence of bigger diameter aggregates, which led to a decrease in coalescence rate and hence foam degradation [29].

#### 3.1.3. PM-EWP Emulsion Activity and Stability

Figure 3 shows the emulsion activity and stability of PM-EWP under various conditions. Increasing the microwave power from 400 to 700 watts lowered the activity and stability of the emulsion considerably at each level of phosphorylation %, and the difference between the different levels of microwave power of 200 and 400 watts was not significant. At each microwave power level, increasing the amount of phosphorylation from 1.5 to 3.5% of emulsion activity displayed a declining tendency, but no significant difference in emulsion stability was observed. Control treatment, i.e., P0-M0, and P0-M700 had the greatest emulsion activity and stability. The control treatment P3.5-M0 and treatments with 3.5% phosphorylation had the lowest emulsion activity and stability, particularly in higher microwave treatments. This might be because the electrostatic repulsion between protein molecules was stronger in the higher microwave treatments, resulting in less protein load at the oil–water interface and less formation of a highly viscoelastic protein membrane [27]. The reduction in emulsifying activity of phosphorylated samples can be attributed to denaturation of the second and third protein structures caused by phosphorylation [26]. According to Soria and Villamiel [30], denatured protein exposes more hydrophobic groups, resulting in protein aggregation and decreased emulsifying function.

#### 3.1.4. PM-EWP Water- and Oil-Holding Capacity

The water-holding capacity of EWP under various conditions is shown in Figure 4. When the phosphorylation level and microwave power were both raised (*p* < 0.05), the water-holding capacity increased. The water-holding capacity of the control sample, P0-M0, was greater than that of the microwave-assisted phosphorylation samples (*p* < 0.05). Figure 4 demonstrates that increasing the phosphorylation (%) at each level of microwave power, as well as increasing the microwave power at each level of phosphorylation reduces oil holding capacity (*p* < 0.05). The oil holding capacity of the control sample, P0-M0, was greater than that of the microwave-assisted phosphorylation samples (*p* < 0.05). The water-holding capacity and wetting behavior were affected by particle size and surface tension, with the smaller particles being less likely to be wetted. The PM-EWP samples exhibited a more porous structure, but the lowest particle size and highest specific surface area [4].

Phosphorylation modification might prevent further exposure of the hydrophobic groups by rearranging them inside the protein molecules [31], which could explain why microwave-assisted phosphorylated treatments have a lower oil-holding capacity and a higher water-holding capacity.

#### 3.1.5. PM-EWP Viscosity Solution 1% (*w*/*v*)

The viscosity of PM-EWP solution 1% (*w*/*v*) under various conditions is shown in Figure 5. The effect of increasing the microwave power to 700 watts on viscosity was not significant (*p* > 0.05). At varying shear rates, the viscosity of egg white power solution in the control group was essentially the same, demonstrating Newtonian fluid properties. All egg white powder solutions following microwave treatment exhibited shear thinning behavior, indicating non-Newtonian fluid properties. This disparity might be explained by the following two factors: first, in the control group, protein molecules with a tight and regular structure were largely dispersed in the same flow layer of solution, and there was no evident entanglement between them. Second, the flexible protein molecules were placed in distinct flow levels of solution and entangled after microwave treatment. This effect was also previously observed by other authors [32]. The apparent viscosity of phosphorylated soy protein isolate and microwave-assisted phosphorylated soy protein isolate reduced to varying degrees. The addition of phosphate groups (${\mathrm{PO}}_{4}^{3-}$) to a protein may alter its molecular structure and surface charge, affecting its hydration and interaction [33].

#### 3.1.6. PM-EWP Phosphorylation Degree

The degree of phosphorylation of PM-EWP samples under various conditions is shown in Figure 6. The degree of phosphorylation decreased significantly when microwave power increased from 200 to 700 watts at each level of phosphorylation. Increasing the proportion of phosphorylation from 1.5 to 3.5% at each level of microwave power resulted in a considerable rise in the degree of phosphorylation. P3.5-M0 had the highest degree of phosphorylation (*p* < 0.05). Dry heating at 2 °C and pH 1 for 2 days in the presence of pyrophosphate enhanced the amount of phosphorus in egg white protein, soluble protein in soybean, ovalbumin, and ovotransferrin, according to Hayashi et al. [34]. In the presence of sodium tripolyphosphate at pH 5.0, 7.0, and 9.0, Xiong and Ma [35] discovered 23, 21, and 18 phosphorylation sites in phosphorylated ovalbumin, including numerous natural phosphorylation sites Ser_68_ and Ser_344_. The list of amino acids that can be phosphorylated has been expanded to include Ser, Thr, Tyr, Arg, and Lys. Since egg white contains naturally phosphorylated proteins such as ovalbumin and egg white riboflavin-binding protein, the phosphorus level of native egg white was 0.6 mg/g. The phosphate group content has grown by more than five times. Moderate microwave treatment alone was shown to boost the phosphorylation levels of egg white proteins, whereas excessive microwave treatment, i.e., 400–600 watts had a detrimental influence on the degree of phosphorylation of the proteins [3].

### 3.2. PM-EWP Physical Properties

#### 3.2.1. PM-EWP Particle Size

Figure 7 shows that the particle-size distribution characteristics of microwave-assisted phosphorylation samples differ significantly (*p* < 0.05). In terms of index d(0.1), the phosphorylation level of 2.5, and 3.5% was more effective in decreasing particle size in lower power microwave samples (*p* < 0.05). At the same time, the phosphorylation level of 1.5% was more effective (*p* < 0.05) in decreasing particle size in microwave samples with higher powers, and the particle size of the control samples (P0-M0 and P0-M700) was lower (*p* < 0.05). In terms of index d(0.5), lower power microwave samples with phosphorylation levels of 3.5 and 2.5% were more effective in lowering particle size (*p* < 0.05). The particle size of the microwave-assisted phosphorylation samples (*p* < 0.05) was smaller than that of the control samples (P0-M0 and P0-M700). In terms of index d(0.9), lower power microwave samples with phosphorylation levels of 2.5, and 3.5% were more effective in reducing particle size (*p* < 0.05). The particle size of the microwave-assisted phosphorylation samples (*p* < 0.05) was smaller than that of the control sample (P0-M700). Microwave samples with a higher power and phosphorylation levels of 1.5 and 3.5% showed a stronger effect on reducing surface mean diameter (*p* < 0.05) in terms of index d(3.2) or surface mean diameter. The surface mean diameter of the control samples (P0-M0, P3.5-M0, and P0-M700, respectively) was substantially smaller than the surface mean diameter of the microwave-assisted phosphorylation samples (*p* < 0.05). In terms of index d(4.3) or volume mean diameter, the phosphorylation levels of 2.5 and 3.5% had a greater effect on reducing the volume mean diameter in microwave samples with 200 watts of radiation (*p* < 0.05). When compared to other PM-EWP samples, the P0-M700, and P0-U0 control samples exhibited the smallest mean volume mean diameter (*p* < 0.05). It might be because of the improved interaction between protein molecules following treatment with microwaves of 200 and 400 watts, respectively, which caused the protein molecules to gather easily. Arzeni et al. [36] discovered an increase in particle size for ultrasonic-assisted phosphorylation EWPs, but only throughout the whole range studied, which they attributed to heat aggregation during the ultrasound treatment. However, if extra hydrophobic groups were exposed after microwave-assisted phosphorylation, these groups might interact with each other, increasing protein size due to protein rearrangement into sub-aggregates from hydrophobic and electrostatic interactions. In the current study, higher microwave power (700 watts) resulted in a significant reduction in the protein particle size of many animal and vegetable proteins tested. This result can be explained by the breakdown of non-covalent associative forces such as hydrogen bonding, and hydrophobic and electrostatic interactions that keep lower protein aggregation in solution. Recently, other authors reported similar effect when the effect of microwave assisted phosphorylation on the physicochemistry and rehydration behavior of egg white powder was tested [3,10]. Covalent interactions were formed between the phosphate groups of sodium triphosphate and the -NH2 and -OH groups of mung bean protein (C–O–P and C–N–P bonds). The insertion of phosphate groups, which may establish a significant number of hydrogen bonds with water molecules, may explain the improved functional properties of phosphorylated mung bean protein. Furthermore, the addition of phosphate groups enhanced the electronegativity of the protein system, which improved electrostatic repulsion between protein molecules and allowed them to disperse more efficiently in the solution system [37]. This can be the reason why the particle size of EWPs especially P3.5-M700 has shown a significant decrease with the increase in microwave power and the percentage of phosphorylation.

#### 3.2.2. PM-EWP Fourier Transform Infrared Spectroscopy (FTIR)

The FTIR spectra of microwave-assisted phosphorylation EWPs freeze-dried, as well as control samples, is shown in Figure 8, with the peaks in the area of wave number 4000–500 cm^−1^ examined. Figure 8A–C compare the effect of different microwave powers on 1.5, 2.5 and 3.5% phosphorylation, respectively. Figure 8D–F compare the effects of different percentages of phosphorylation in different microwave powers of 200, 400 and 700 watts, respectively.

Microwave-assisted phosphorylated samples showed additional adsorption peaks at wave numbers 970–980 cm^−1^ and 890–900 cm^−1^, which were attributed to P-O and P=O stretching, respectively [3]. Tang et al. [38] found that the egg white protein side chain was efficiently added to phosphate groups. The protein’s second structure changed after microwave treatment, and the beta-sheet expanded somewhat. Because Wang and Chi [32,33] found that microwave-assisted phosphorylation of soy bean isolate resulted in a substantial increase in beta sheet and beta turn, but a significant decrease in alpha helix and random coil. This could be due to an increase in the electronegativity of the phosphorylated protein system, which increased the electrostatic repulsion between the protein molecules, resulting in the formation of new hydrogen bonds between non-polar groups of the protein, and as a result, the content of beta sheet increased while that of random coil decreased. This demonstrated that following phosphorylation, the protein structure became more compact and the ordering improved. The amide I band (1600–1700 cm^−1^) and the amide II band (1480–1575 cm^−1^) are two absorption bands in the infrared region of proteins that may be used to study secondary structures [39]. The –NH and –CN stretching were assigned wave numbers of 3298 cm^−1^ and 1397 cm^−1^, respectively. Microwave-assisted phosphorylation EWPs showed a 2 cm^−1^ red shift at –NH stretching and a 2 cm^−1^ blue shift at –CN stretching, compared to the P0-M0 sample, while microwaved-EWP only had a 1 cm^−1^ blue shift at –CN stretching. In general, microwave-assisted phosphorylation alterations can lead to further structural modifications in proteins [3,10].

#### 3.2.3. PM-EWP Scanning Electron Microscopy (SEM)

In comparison to the control sample (P0-M0), the particle size of microwave-assisted phosphorylated EWPs was decreased (Figure 9). The PM-EWP samples had a finer and more porous structure, but the smallest particle size and largest specific surface area. As a result, the hydrophilicity of PM-EWP particles was lower than that of control powders. In general, structural changes in proteins generated by microwave-induced strong heat denaturation, as well as electrical repulsion created by negative electric charges on phosphorus groups caused by phosphorylation, may be justified [22,23].

**Figure 8 foods-11-02711-f008:**
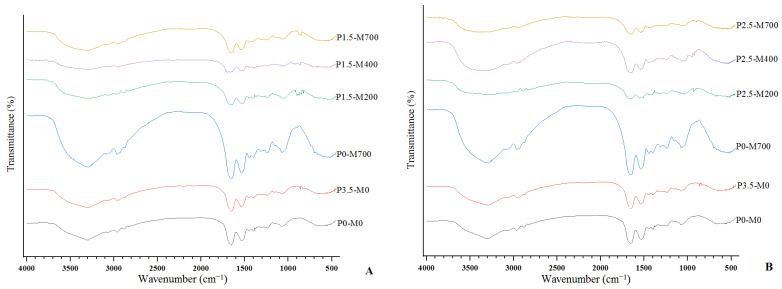
Fourier transform infrared spectroscopy (FTIR) of egg white powder treated with different phosphorylation (%) and microwave powers (W) (**A**–**C**) compare the effect of different microwave powers on 1.5, 2.5, and 3.5% phosphorylation, respectively. (**D**–**F**) compare the effects of different percentages of phosphorylation in microwave powers of 200, 400 and 700 watts, respectively.

## 4. Conclusions

In the treatments that used phosphorylation and microwave simultaneously, P1.5-M200 has the highest oil-holding capacity, emulsion stability, and emulsion activity, and P2.5-M200 has the highest foam capacity, P2.5-M400 has the largest particle size, and P3.5-M200 has the highest solution viscosity by 1% (*w*/*v*), water-holding capacity, and foam stability. The binding of negatively charged phosphate groups to the serine, threonine, and tyrosine side chains was discovered by FTIR spectroscopy of the unfolding structure of egg white protein. This binding also caused alterations in the protein’s secondary structure. The phosphorylation and microwave treatments, according to SEM tests, reduced the particle size of the egg white protein powder while increasing the surface area of the protein molecules. According to the findings, microwave-assisted phosphorylation modification followed by freeze drying is a practical approach to enhance the functional qualities of egg white protein. In order to enhance the functional properties of EWP, coupled phosphorylation with non-thermal approaches (ultrasound, pulsed electric field, high-pressure processing, radio frequency, ultraviolet light, and cold plasma) is suggested as a direction for future study.

## Figures and Tables

**Figure 1 foods-11-02711-f001:**
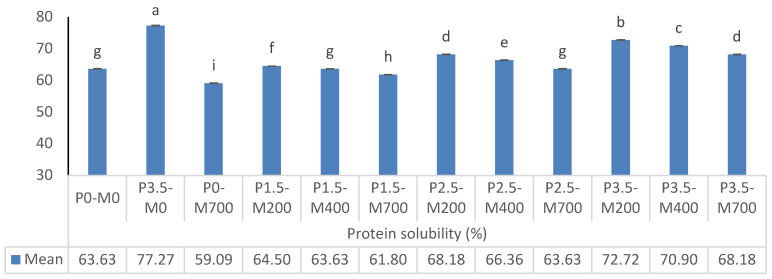
The effect of phosphorylation (%) and microwave power (W) on protein solubility of egg white powder. Means (*n* = 3) with different letters in each column indicate significant difference (*p* < 0.05).

**Figure 2 foods-11-02711-f002:**
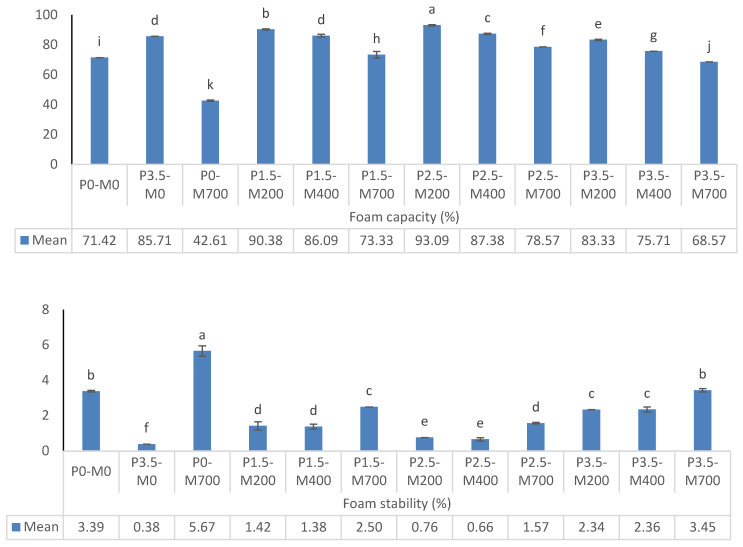
The effect of phosphorylation (%) and microwave power (W) on foam capacity and stability of egg white powder. Means (*n* = 3) with different letters in each column indicate significant difference (*p* < 0.05).

**Figure 3 foods-11-02711-f003:**
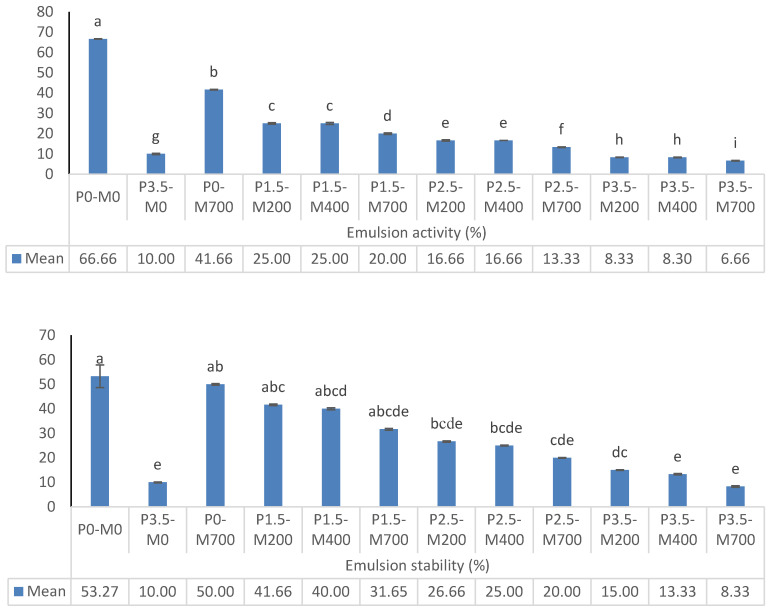
The effect of phosphorylation (%) and microwave power (W) on emulsion activity and stability of egg white powder. Means (*n* = 3) with different letters in each column indicate significant difference (*p* < 0.05).

**Figure 4 foods-11-02711-f004:**
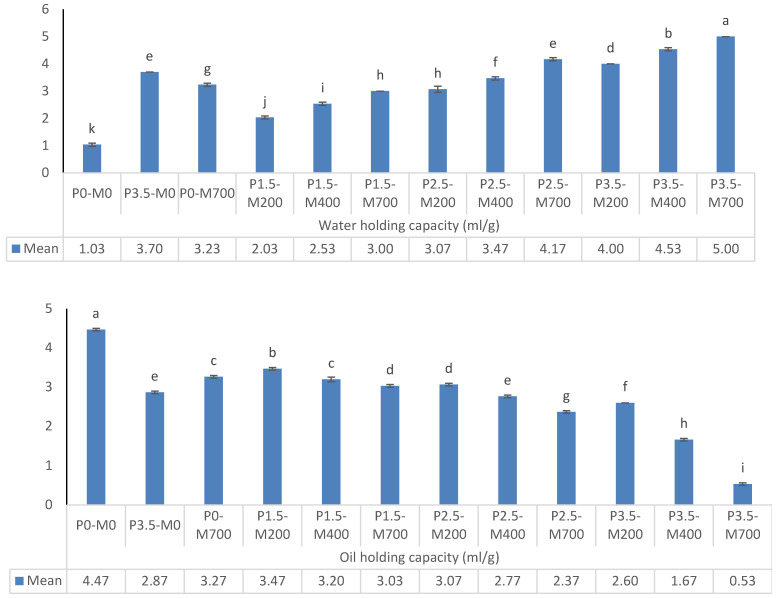
The effect of phosphorylation (%) and microwave power (W) on water-holding capacity and oil holding capacity of egg white powder. Means (*n* = 3) with different letters in each column indicate significant difference (*p* < 0.05).

**Figure 5 foods-11-02711-f005:**
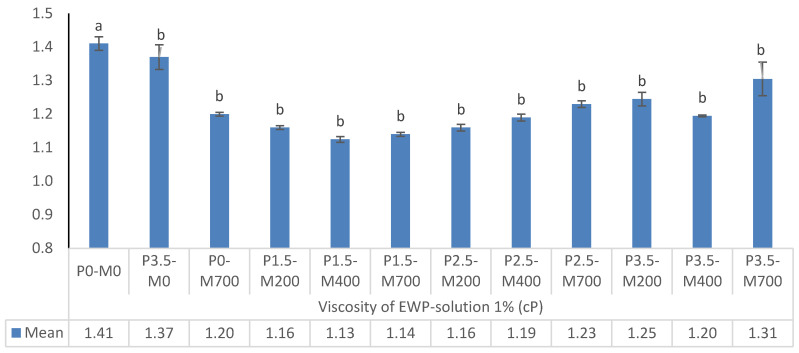
The effect of phosphorylation (%) and microwave power (W) on viscosity of egg white powder-solution 1%. Means (*n* = 3) with different letters in each column indicate significant difference (*p* < 0.05).

**Figure 6 foods-11-02711-f006:**
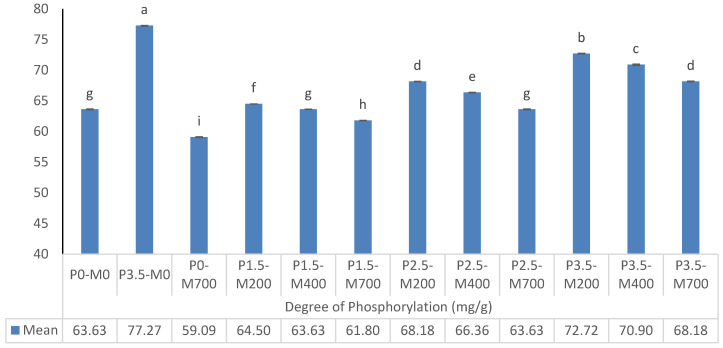
The effect of phosphorylation (%) and microwave power (W) on degree of phosphorylation of egg white powder. Means (*n* = 3) with different letters in each column indicate significant difference (*p* < 0.05).

**Figure 7 foods-11-02711-f007:**
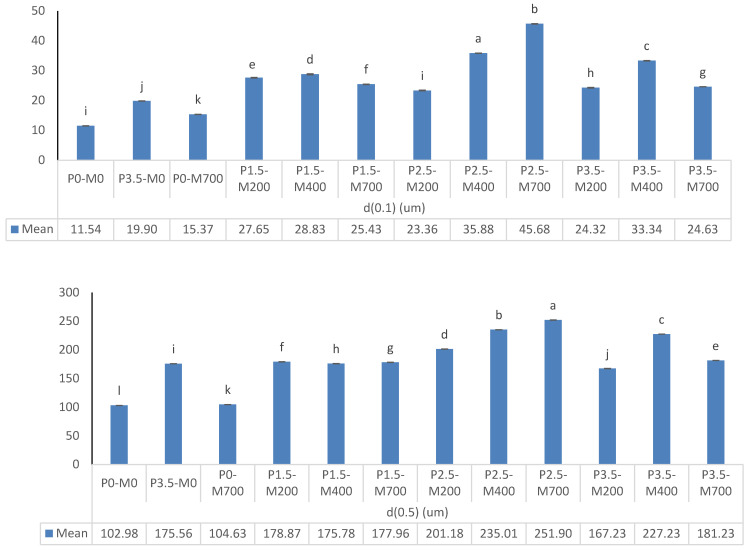
The effect of phosphorylation (%) and microwave power (W) on particle size distribution of egg white powder. Means (*n* = 3) with different letters in each column indicate significant difference (*p* < 0.05).

**Figure 9 foods-11-02711-f009:**
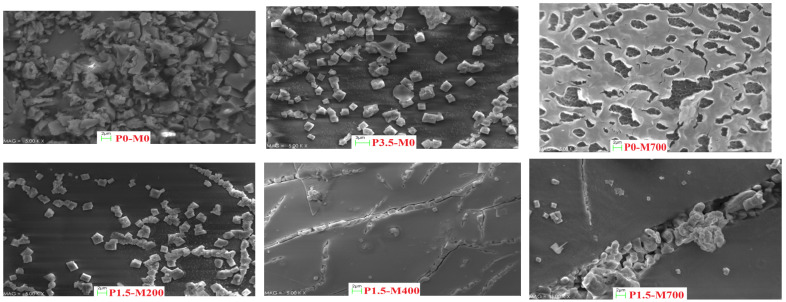
Scanning electron microscopy (SEM) of egg white powder treated with different phosphorylation (%) and microwave powers (W).

## Data Availability

The data presented in this study are available on request from the corresponding author.

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
