# Peer review of "The Effects of Phosphorylation and Microwave Treatment on the Functional Characteristics of Freeze-Dried Egg White Powder"

_foods, 2022, doi:10.3390/foods11172711_

Round 1
Reviewer 1 Report
1. L15 - Check the sentence "by the freeze-drying" - after checking the methods, the authors have freeze-dried the sample after it has been phosphorylated and microwaved, but this sentence provides a different meaning; please check.
2. Throughout the paper author used "percentage" change to %.
3. Please include microwave duration in the abstract.
4. L16-17 - "were investigated ..... level" - remove this sentence.
5. L18-20 - author gave the information that the treated samples had shown increased tested parameters, but however, there many fluctuated and not consistent; if the author wants to give significant results, then better use one tested treatment which is good and shown in the abstract.
6. Microwave-assisted phosphorylation is the wrong term; please provide keywords.
7. Abbreviation, and practical application, are not the format style of foods journal; please check.
8. L54 - has cheaper packaging cost - please remove these phrases. "easier to store" - mention the temperature in the brackets.
9. Authors have used many "because" words throughout the manuscript; I think it's overdone; please reduce it if possible.
10. In the introduction, the authors provided very few details about microwave and freeze dry related to egg white; authors should specify more clearly and support the author's hypothesis with previously published papers.
11. Author claims another type of method is very expensive but is the author's experimental design an inexpensive practice?
12. L91 - what type of egg sources such as chicken, duck, or quail should report with specie name?
13. L95 - reagents from analytical grade quality - please provide sourcing details such as brand name or company details etc.
14. L107 - author mentioned citric acid dihydrate, but no concentration was provided; please provide.
15. L110 - resuspended in cold water for how long, when finished, what is the final temperature, should include as well.
16. L229 - the author provides an abbreviation for samples "P3.5-M0," but nowhere is given in abb form before; please provide all the samples with an abbreviation if the author intended to use them in the material and method section.
17. L227-228 - Confusing phrased; please rewrite.
18. L235 - in a small amount of protein - is the author mean about the quantity or size of protein?
19. I recommend authors change the results from table form to figure form; it is more easier to compare and observe the trend, but at present, it is congested and too difficult to track and check.
20. L281-282- In these phrases, I couldn't be able to get what the authors were trying to explain; please rewrite for better understanding.
21. L292 - According to table 1 - Please remove these phrases, as the authors have already provided in the previous line whereabout the result shown, again repeating is not needed, and this point is matched with other resutls as well, as an author has given this sentence, in many places. please revise it accordingly.
22. L330 - possibly due to a more......distribution - How do authors conclude on this point? in the emulsification results, the author mentioned denaturation increased with microwave and phosphorylation; if the denature is more, then uniformity is not possible in the protein structure. please explain.
23. L332-334 - Please change it., not relevant.
24. L352 - PO4 3- -please provide the full form and quote the abbreviation.
25. Section 3.2.1 - please add more discussion for this section and provide more references to support the author's findings.
26. L430-431- The protein......somewhat. Please provide evidence for this phenomenon.
27. section 3.2.3 - I recommend a complete rewrite; moreover, the authors have not provided the results for SEM; without the results, reviewing this section is impossible.
28. In conclusion - "according to ....PM-EWP". It is a reputation from the abstract, please rewrite and provide significant findings only, not needed to mention all.
29. "The phosphorylation .... to SEM analysis - how could authors find these from SEM results? Please explain.
*****
Author Response
Dear Prof.
We used reviewers and your insightful comments and we corrected parts of the article which were not meaningful (see following text) and the manuscript was entirely corrected according to the reviewer viewpoints. Please do not hesitate to ask us any questions about the submitted manuscript. This research, like many other scientific studies, has many weaknesses. I would like to express my deep gratitude to the hard-working referee of the Journal for his English language re-editing and wise-scientific re-judgment.
Sincerely yours,
Mohammad Goli
Hi dear editorial board and reviewer 1
Yours sincerely, thank you, the editor and reviewer 1, for your kind consideration for your scientific attention to my manuscript, for having read it, and for your valuable scientific and intelligent comments. I hope that by using your guidance, dear referees will attempt to refine the article and raise its scientific level. The yellow, green, and turquoise highlight in the uncleaned-revised manuscript related to the final reviewer 1, 2, and 3 proposed amendments, respectively.
- L15 - Check the sentence "by the freeze-drying" - after checking the methods, the authors have freeze-dried the sample after it has been phosphorylated and microwaved, but this sentence provides a different meaning; please check.
- I am really pleased with your precision. As you mentioned and understood, the treatments of phosphorylation and microwave before the freeze-drying operation were employed on the liquid egg white, thus we used the pre-treatment term in the text so that the concept of our study could be clearly explained.
- Throughout the paper author used "percentage" change to %.
- Thank you for your very subtle and intelligent comments. All your suggestions (percentage words) in the text were converted to %
- Please include microwave duration in the abstract.
- Your scientific suggestion was done as the follow:
The effects of phosphorylation pre-treatments at 1.5, 2.5, and 3.5% levels, as well as microwave application at 200, 400, and 700 watts levels for two min, on the functional parameters of egg white powder obtained by the freeze dryer procedure were investigated.
- L16-17 - "were investigated ..... level" - remove this sentence.
- Your intelligent comment was done as the following text:
- The effects of phosphorylation pre-treatments at 1.5, 2.5, and 3.5% levels, as well as microwave application at 200, 400, and 700 watts levels for two min, on the functional parameters of egg white powder obtained by the freeze dryer procedure were investigated.
- L18-20 - author gave the information that the treated samples had shown increased tested parameters, but however, there many fluctuated and not consistent; if the author wants to give significant results, then better use one tested treatment which is good and shown in the abstract.
- Thank you so much for the best scientific comment. I amended it according to your suggestion as the following:
P1.5-M200 has the highest oil holding capacity, emulsion stability, and emulsion activity, and P2.5-M200 has the highest foam capacity, P2.5-M400 has the largest particle size, and P3.5-M200 has the highest degree of phosphorylation and protein solubility, and P3.5-M200 has the highest solution viscosity by 1% (w/v), water holding capacity, and foam stability, In the treatments that used phosphorylation and microwave simultaneously.
- Microwave-assisted phosphorylation is the wrong term; please provide keywords.
- Keywords: Phosphorylation-microwave treatment; Egg-white protein powder; Physicochemical characterization; Particle-size distribution; Scanning electron microscopy; Fourier transform infrared spectroscopy
- Abbreviation, and practical application, are not the format style of foods journal; please check.
- Thank you so much and I omit these two section pointed to them.
- L54 - has cheaper packaging cost - please remove these phrases. "easier to store" - mention the temperature in the brackets.
- Your kindly comment was done as the follow:
Egg white powder has become a popular commercial product as an alternative for fresh and liquid eggs because it can avoid microbiological and oxidative deterioration and has cheaper packaging costs [2].
- Authors have used many "because" words throughout the manuscript; I think it's overdone; please reduce it if possible.
- In the whole text of the article, “because” word was used 10 times, which was reduced to 3 times with your valuable comment, dear referee. Please pay attention to an example of this amendment.
Egg white contains a variety of proteins, including ovalbumin, ovotransferrin, ovomucoid, ovomucin, and lysozyme [1]. Due to its high nutritional content and good functional characteristics, egg white is an important component in the food industry. Egg white powder has become a popular commercial product as an alternative for fresh and liquid eggs because it can avoid microbiological and oxidative deterioration and has cheaper packaging costs [2]. Proteins' functional properties can be altered by physical, chemical, and enzymatic treatments. Since chemicals are hazardous to human health, the use of chemical techniques to modify proteins in the food industry is strictly prohibited [3]. The use of enzyme modification is often restricted due to the high cost of enzymes as well as difficult and complex operating conditions. As a result, physical protein modification (thermal processing, i.e., microwave treatment) is in considerable interest [4].
- In the introduction, the authors provided very few details about microwave and freeze dry related to egg white; authors should specify more clearly and support the author's hypothesis with previously published papers.
- Thank you so much for wise and scientific comment resulted in the better conception of introduction section. We amended introduction according your suggestion as the following text:
For a few years, microwave-assisted chemical procedures have been employed to achieve chemical reactions. When compared to traditional heating techniques, chemical synthesis with microwave help can significantly shorten reaction duration while also improving yield and product purity. Microwave-assisted approaches have previously been used by certain researchers in the chemical alteration of dietary protein [7]. When compared to the traditional dry-heating approach, the microwave treatment can significantly shorten reaction times and speed the phosphorylation process. Li et al. [2] used a spray dryer to produce three different egg white protein powders (untreated, microwave-alone treated, and microwave-assisted phosphorylation modification by sodium tri-phosphate (STP or Na3PO4)) and investigated their physicochemistry and rehydration behavior.
There are several drying techniques available for producing sodium tri-phosphate egg white powder (STP-EWP), and these drying methods should be chosen based on drying efficiency and dried product quality. It has been found that freeze-drying proteins retain their natural structural shape better because they are subjected to fewer heat and water evaporation-related stresses. The freeze-dried particles are more porous and shrink less. According to Lili et al. [8] freeze-dried STP-EWP powders outperformed spray-dried STP-EWP powders in terms of solubility, emulsion stability, water holding capacity, oil and water absorption capacity, and heat gel strength. Freeze drying was found to be the most effective way for producing modified EWP powders with enhanced functional characteristics.
- Author claims another type of method is very expensive but is the author's experimental design an inexpensive practice?
- Thank you so much for intelligent comment and I amended the expression of our justify as the following text:
The use of enzyme modification is often restricted due to the high cost of enzymes as well as difficult and complex operating conditions.
- L91 - what type of egg sources such as chicken, duck, or quail should report with specie name?
- The intelligent comment was done as the follow:
Fresh chicken eggs were provided from a local farm (Simorgh LTD, Isfahan, Iran). The analytical grade ingredients were disodium hydrogen phosphate (DSP, Na2HPO4), citric acid dihydrate, nitric acid, sulfuric acid, perchloric acid, trichloroacetic acid, potassium bromide, bovine serum albumin, and the rest of the reagents (Merck, Germany).
- L95 - reagents from analytical grade quality - please provide sourcing details such as brand name or company details etc.
- Thank you so much for wise comment and I amended it as the follow:
Fresh chicken eggs were provided from a local farm (Simorgh LTD, Isfahan, Iran). The analytical grade ingredients were disodium hydrogen phosphate (DSP, Na2HPO4), citric acid dihydrate, nitric acid, sulfuric acid, perchloric acid, trichloroacetic acid, potassium bromide, bovine serum albumin, and the rest of the reagents (Merck, Germany).
- L107 - author mentioned citric acid dihydrate, but no concentration was provided; please provide.
- Your suggestion was done in the manuscript text as th following:
DSP was dissolved in the egg-white protein solution to generate 15, 25, and 35 gL-1 mixed solutions (i.e. 1.5, 2.5, and 3.5 % phosphorylated treatments) at pH 8 by lowering the pH with citric acid dihydrate (0.033 M, pH 2.2).
- L110 - resuspended in cold water for how long, when finished, what is the final temperature, should include as well.
- It was done according to the wise comment as the follow:
Then the solutions were microwaved for 2 min with powers of 200, 400 and 700 watts. The solutions were immediately suspended in cold water at 0 °C for 4-5 min in order to prevent excessive reactivity and bring the temperature of the solution up to 10-15 °C.
- L229 - the author provides an abbreviation for samples "P3.5-M0," but nowhere is given in abb form before; please provide all the samples with an abbreviation if the author intended to use them in the material and method section.
- In materials and methods I pointed it as the follow:
2.3. Preparation of microwave-assisted phosphorylated egg-white powder
DSP was dissolved in the egg-white protein solution to generate 15, 25, and 35 gL-1 mixed solutions (i.e. 1.5, 2.5, and 3.5 % phosphorylated treatments) at pH 8 by lowering the pH with citric acid dihydrate (0.033 M, pH 2.2). Then the solutions were microwaved for 2 min with powers of 200, 400 and 700 watts. The solutions were immediately suspended in cold water at 0 °C for 4-5 min in order to prevent excessive reactivity and bring the temperature of the solution up to 10-15 °C. The extra phosphates were then precipitated as calcium phosphate by adding 19, 32, and 44 gL-1 of calcium chloride to 1.5, 2.5, and 3.5 % phosphorylated solutions, respectively. To remove excess calcium phosphate, the solutions were centrifuged at 3000 g for 5 minutes at room temperature. At a temperature of -50 °C and an air pressure of 0.04-0.06 mbar, the supernatant was dried in a freeze drier [2]. The control treatment, also known as P0-M0, refers for no phosphorylation and microwave treatment. Other treatments, such as P3.5-M700, indicate for 3.5% phosphorylation and 700 watts of microwave.
- L227-228 - Confusing phrased; please rewrite.
Thank you for your wise comment and I amended it according to the follow:
Table 1 shows that protein solubility decreased substantially as microwave power increased from 200 to 700 watts at each level of phosphorylation (%) but significantly increased at each level of microwave power when phosphorylation level increased from 1.5 to 3.5%.
- L235 - in a small amount of protein - is the author mean about the quantity or size of protein?
- I mean is based on quantity of protein. And the sentence pointed was corrected according to the following:
This phenomenon may be explained by the fact that microwave radiation significantly affected a protein's solubility when the quantity of phosphate groups in the protein was low.
- I recommend authors change the results from table form to figure form; it is more easier to compare and observe the trend, but at present, it is congested and too difficult to track and check.
- Considering that the number of figures resulting from the two tables proposed by you, dear referee, would be very high and with the previous figures, a total of 16 figures would be created, and that there is usually a limit of figures and tables in authoritative journals, and mostly the maximum number of figures and Tables, it should not be more than 6. Therefore, please be satisfied with the current situation so that I do not face the problems of the editor of the journal in the future
- L281-282- In these phrases, I couldn't be able to get what the authors were trying to explain; please rewrite for better understanding.
- It was corrected to get better explain and rewrite it again as the follow:
At all the same levels of applied microwave power, 3.5% and 2.5% phosphorylation showed the highest and lowest foam stability, respectively.
- L292 - According to table 1 - Please remove these phrases, as the authors have already provided in the previous line whereabout the result shown, again repeating is not needed, and this point is matched with other resutls as well, as an author has given this sentence, in many places. please revise it accordingly.
- I I tried to remove the term " According to Table 1 or 2" in the entire text of the article, for example, you can see one of the changed items in the text below.
Increasing the microwave power from 400 to 700 watts greatly enhanced the stability of the foam at each level of phosphorylation %, although the difference between different levels of microwave power of 200 and 400 watts was not significant.
- L330 - possibly due to a more......distribution - How do authors conclude on this point? in the emulsification results, the author mentioned denaturation increased with microwave and phosphorylation; if the denature is more, then uniformity is not possible in the protein structure. please explain.
- Your point of view is truly appreciated, and in order to avoid misleading the article's reader, I attempted to delete this statement from the text.
Pore size and pore volume measurements revealed that the PM-EWP powder became more porous, possibly due to a more uniform particle distribution. Finally, the particle size of PM-EWP decreased, and the material's structure became more porous (Li et al., 2020).
- L332-334 - Please change it., not relevant.
- Thank you due to wise and intelligent comment for better interpretation of results, therefore the following text which is not relevant was deleted.
- Pore size and pore volume measurements revealed that the PM-EWP powder became more porous, possibly due to a more uniform particle distribution. Finally, the particle size of PM-EWP decreased, and the material's structure became more porous (Li et al., 2020).
- L352 - PO4 3- -please provide the full form and quote the abbreviation.
- It was expressed as the full form and quote the abbreviation in parenthesis as the following text:
The addition of phosphate groups ( ) to a protein may alter its molecular structure and surface charge, affecting its hydration and interaction [31].
- Section 3.2.1 - please add more discussion for this section and provide more references to support the author's findings.
- Thank you so much for wise consideration. The following text was added to section 3.2.1 for supporting our finding.
Covalent interactions were formed between the phosphate groups of sodium triphosphate and the -NH2 and -OH groups of mung bean protein (C–O–P and C–N–P bounds). The insertion of phosphate groups, which may establish a significant number of hydrogen bonds with water molecules, may explain the improved functional properties of phosphorylated mung bean protein. Furthermore, the addition of phosphate groups enhanced the electronegativity of the protein system, which improved electrostatic repulsion between protein molecules and allowed them to disperse more efficiently in the solution system [35]. This can be the reason why the particle size of EWPs especially P3.5-M700 has shown a significant decrease with the increase in microwave power and the percentage of phosphorylation.
- L430-431- The protein......somewhat. Please provide evidence for this phenomenon.
- Thank you so much for your scientific suggestion and I provided one evidence as the following:
The protein's second structure changed after microwave treatment, and the beta-sheet expanded somewhat. Because Wang and Chi [31] found that microwave-assisted phosphorylation of soy bean isolate resulted in a substantial increase in beta sheet and beta turn, but a significant decrease in alpha helix and random coil. This could be due to an increase in the electronegativity of the phosphorylated protein system, which increased the electrostatic repulsion between the protein molecules, resulting in the formation of new hydrogen bonds between non-polar groups of the protein, and as a result, the content of beta sheet increased while that of random coil decreased. This demonstrated that following phosphorylation, the protein structure became more compact and the ordering improved.
- section 3.2.3 - I recommend a complete rewrite; moreover, the authors have not provided the results for SEM; without the results, reviewing this section is impossible.
- Unfortunately, I see that Figure 2, which was already submitted to the article, but it was not visible to your uploading manuscript file. This figure was included in “ response to reviewer-1 file” and of course it was included in the revised final file.
- In conclusion - "according to ....PM-EWP". It is a reputation from the abstract, please rewrite and provide significant findings only, not needed to mention all.
- Thank you so much for wise consideration. I amended conclusion as a new approach according to your scientific comment as the follow:
In the treatments that used phosphorylation and microwave simultaneously, P1.5-M200 has the highest oil holding capacity, emulsion stability, and emulsion activity, and P2.5-M200 has the highest foam capacity, P2.5-M400 has the largest particle size, and P3.5-M200 has the highest solution viscosity by 1% (w/v), water holding capacity, and foam stability. The binding of negatively charged phosphate groups to the serine, threonine, and tyrosine side chains was discovered by FTIR spectroscopy of the unfolding structure of egg white protein. This binding also caused alterations in the protein's secondary structure. The phosphorylation and microwave treatments, according to SEM tests, reduced the particle size of the egg white protein powder while increasing the surface area of the protein molecules. According to the findings, microwave-assisted phosphorylation modification followed by freeze drying is a practical approach to enhance the functional qualities of egg white protein. In order to enhance the functional properties of EWP, coupled phosphorylation with non-thermal approaches (ultrasound, pulsed electric field, high-pressure processing, radio frequency, ultraviolet light, and cold plasma) is suggested as a direction for future study.
- "The phosphorylation .... to SEM analysis - how could authors find these from SEM results? Please explain.
- Unfortunately, I see that Figure 2, which was submitted in the article, was already sent, but it was not visible to you, dear referee. This figure was included in “ response to reviewer-1 file” and of course it was included in the revised final file.
|
|
|
|
|
|
|
|
|
|
|
|
|
|
|
|
|
Fig 2. Scanning electron microscopy (SEM) of egg white powder treated with different phosphorylation (%) and microwave powers (W) |
||
|
|
||

Reviewer 2 Report
Interesting publication. No similar paper was published so far. English level is acceptable.
Some important papers are missing:
Lili, Liu et al. “Effects of freeze-drying and spray drying processes on functional properties of phosphorylation of egg white protein.” International Journal of Agricultural and Biological Engineering 8 (2015): 116-123.
Egg white protein (EWP) was phosphorylated with Sodium Tripolyphosphate (STP) at pH 4.5. Freeze drying and spray drying were used for drying purpose and the effects of these drying methods on the functional properties were investigated. The functional properties of native and modified proteins were also determined. The results demonstrated that phosphorylation of EWP markedly improved its functional properties, and that it was more effective for the food industry. The freeze-dried STP-EWP powders were superior in terms of solubility, emulsion stability, water holding capacity, oil and water absorption capacity and heat gel strength than spray-dried STP-EWP powders. The results in viscosity showed no significant differences between freeze-dried and spray dried. The spray-dried powders were better in terms of foaming ability and foam stability than freeze-dried powders. However, the spray drying required the longest time to produce. Freeze drying was found to be the best method in terms of production of modified EWP powders with superior functional properties.
Muhammad Talha Afraz, et al. "Impact of Novel Processing Techniques On the Functional Properties of Egg Products and Derivatives: A Review." Journal of food process engineering, v. 43 ,.12 pp. e13568. doi: 10.1111/jfpe.13568
Thermal processing employed for stabilizing and improving shelf‐life of egg components is known to have adverse effect on heat‐sensitive proteins leading to protein denaturation and aggregation thus, reducing the required functional, technological, and overall quality of egg proteins and other constituents. Therefore, the current challenge is to identify novel processing techniques that not only improve the intrinsic functional properties of eggs or its components, but also improve the quality of the product. This review focuses on the use of technologies such as ultrasound, pulsed electric field, high‐pressure processing, radiofrequency, ultraviolet light, microwave, and cold plasma for egg products. These novel technologies are known for their advantages over thermal treatments especially in protecting the heat sensitive nature and retaining the overall quality of the egg and egg products. Availability of alternatives processing has significantly improved the structural properties, techno‐functional, nutritional and as well improving the safety egg and egg products.
86 correct
91 correct
254-273 can be moved to the Introduction
341 correct
4. Conclusion....You repeated exactly the Abstract. Rewrite it and make real conclusions with a perspective for future research.
Author Response
Dear Prof.
We used reviewers and your insightful comments and we corrected parts of the article which were not meaningful (see following text) and the manuscript was entirely corrected according to the reviewer viewpoints. Please do not hesitate to ask us any questions about the submitted manuscript. This research, like many other scientific studies, has many weaknesses. I would like to express my deep gratitude to the hard-working referee of the Journal for his English language re-editing and wise-scientific re-judgment.
Sincerely yours,
Mohammad Goli
Hi dear editorial board and reviewer 2
Yours sincerely, thank you, the editor and reviewer 2, for your kind consideration for your scientific attention to my manuscript, for having read it, and for your valuable scientific and intelligent comments. I hope that by using your guidance, dear referees will attempt to refine the article and raise its scientific level. The yellow, green, and turquoise highlight in the uncleaned-revised manuscript related to the final reviewer 1, 2, and 3 proposed amendments, respectively.
Interesting publication. No similar paper was published so far. English level is acceptable.
- Thank you so much for your intelligent and scientific comments, which resulted in a better understanding of the manuscript.
Some important papers are missing:
Lili, Liu et al. “Effects of freeze-drying and spray drying processes on functional properties of phosphorylation of egg white protein.” International Journal of Agricultural and Biological Engineering 8 (2015): 116-123.
Egg white protein (EWP) was phosphorylated with Sodium Tripolyphosphate (STP) at pH 4.5. Freeze drying and spray drying were used for drying purpose and the effects of these drying methods on the functional properties were investigated. The functional properties of native and modified proteins were also determined. The results demonstrated that phosphorylation of EWP markedly improved its functional properties, and that it was more effective for the food industry. The freeze-dried STP-EWP powders were superior in terms of solubility, emulsion stability, water holding capacity, oil and water absorption capacity and heat gel strength than spray-dried STP-EWP powders. The results in viscosity showed no significant differences between freeze-dried and spray dried. The spray-dried powders were better in terms of foaming ability and foam stability than freeze-dried powders. However, the spray drying required the longest time to produce. Freeze drying was found to be the best method in terms of production of modified EWP powders with superior functional properties.
- Thank you so much for wise and scientific comment resulted in the better conception of introduction section. We amended introduction according your suggestion as the following text:
There are several drying techniques available for producing sodium tri-phosphate egg white powder (STP-EWP), and these drying methods should be chosen based on drying efficiency and dried product quality. It has been found that freeze-drying proteins retain their natural structural shape better because they are subjected to fewer heat and water evaporation-related stresses. The freeze-dried particles are more porous and shrink less. According to Lili et al. [8] freeze-dried STP-EWP powders outperformed spray-dried STP-EWP powders in terms of solubility, emulsion stability, water holding capacity, oil and water absorption capacity, and heat gel strength. Freeze drying was found to be the most effective way for producing modified EWP powders with enhanced functional characteristics.
Muhammad Talha Afraz, et al. "Impact of Novel Processing Techniques On the Functional Properties of Egg Products and Derivatives: A Review." Journal of food process engineering, v. 43 ,.12 pp. e13568. doi: 10.1111/jfpe.13568
Thermal processing employed for stabilizing and improving shelf‐life of egg components is known to have adverse effect on heat‐sensitive proteins leading to protein denaturation and aggregation thus, reducing the required functional, technological, and overall quality of egg proteins and other constituents. Therefore, the current challenge is to identify novel processing techniques that not only improve the intrinsic functional properties of eggs or its components, but also improve the quality of the product. This review focuses on the use of technologies such as ultrasound, pulsed electric field, high‐pressure processing, radiofrequency, ultraviolet light, microwave, and cold plasma for egg products. These novel technologies are known for their advantages over thermal treatments especially in protecting the heat sensitive nature and retaining the overall quality of the egg and egg products. Availability of alternatives processing has significantly improved the structural properties, techno‐functional, nutritional and as well improving the safety egg and egg products.
- Thank you so much for intelligent comment which resulted in the better conception of introduction section. We amended introduction according your suggestion as the following text:
The current challenge is to identify novel processing techniques (such as ultrasound, pulsed electric field, high pressure processing, radio frequency, ultraviolet light, microwave, and cold plasma for egg products) that improve not only the intrinsic functional properties of eggs or their components, but also the product's quality. These innovative methods are recognized for their advantages over thermal treatments, particularly in protecting the heat sensitive nature of eggs and egg products. The availability of alternative processing methods has substantially increased the structural features, technological functionality, nutritional value, and safety of eggs and egg products [5].
86 correct
- Thank you so much for your wise suggestion. I amended that sentence according to the following text:
For the first time, the combined effects of phosphorylation with Na2HPO4 (1.5, 2.5, and 3.5 %), microwave treatment with power (200, 400, and 700 watt), and finally freeze drying (0.04-0.06 mbar & -50 ) on the functional characteristics including protein solubility, foam capacity, foam stability, emulsion activity, emulsion stability, water holding capacity , oil holding capacity, viscosity of solution 1%, as well as degree of phosphorylation, particle-size distribution, FTIR phosphorylation analysis, and SEM morphological analysis of egg white protein powder were investigated.
91 correct
- Thank you so much for your wise suggestion. I amended that sentence according to the following text:
Fresh chicken eggs were provided from a local farm (Simorgh LTD, Isfahan, Iran).
254-273 can be moved to the Introduction
- Lines 254 to 273 have been used as discussion foaming ability and stability, but for your satisfaction, respected and scientific referee, I improved the topic of the introduction in a favorable way.
341 correct
- Thank you so much for your wise suggestion. I amended that sentence according to the following text:
3.1.5. PM-EWP viscosity solution 1% (w/v)
The viscosity of PM-EWP solution 1% (w/v) under various conditions is shown in Table 1.
- Conclusion....You repeated exactly the Abstract. Rewrite it and make real conclusions with a perspective for future research.
- Thank you so much for your intelligent comment. I concise and rewrite conclusion as you suggested and the end of the conclusion pointed to the perspective for future research.
In the treatments that used phosphorylation and microwave simultaneously, P1.5-M200 has the highest oil holding capacity, emulsion stability, and emulsion activity, and P2.5-M200 has the highest foam capacity, P2.5-M400 has the largest particle size, and P3.5-M200 has the highest solution viscosity by 1% (w/v), water holding capacity, and foam stability. The binding of negatively charged phosphate groups to the serine, threonine, and tyrosine side chains was discovered by FTIR spectroscopy of the unfolding structure of egg white protein. This binding also caused alterations in the protein's secondary structure. The phosphorylation and microwave treatments, according to SEM tests, reduced the particle size of the egg white protein powder while increasing the surface area of the protein molecules. According to the findings, microwave-assisted phosphorylation modification followed by freeze drying is a practical approach to enhance the functional qualities of egg white protein. In order to enhance the functional properties of EWP, coupled phosphorylation with non-thermal approaches (ultrasound, pulsed electric field, high-pressure processing, radio frequency, ultraviolet light, and cold plasma) is suggested as a direction for future study.

Reviewer 3 Report
Dear Authors,
The Current paper that you submitted to the Foods Journal it is interesting to the readers, and it is significant to the field, however, in this form, it's very messy, hard to follow, and is not prepared as per journal recommendations.
The paper is extremely poor presented. Introduction chapter do not present sufficient relevant background information.
In the entire paper, use % the symbol instead of percent.
In the text, reference numbers should be placed in square brackets [ ], and placed before the punctuation; for example [1], [1–3] or [1,3]. For embedded citations in the text with pagination, use both parentheses and brackets to indicate the reference number and page numbers; for example [5] (p. 10). or [6] (pp. 101–105).
The results are not very clear presented, and all the information is presented in only two tables, and is very hard to understand it. I suggest to split the tables into smallest ones, which are much more clear and logical.
The authors mention in the paper Figure 2, but this figure is missing from the paper.
The discussions are not sufficient.
Figure 1, is not understandable. Is poorly presented, please replace it with another one which is much more clear.
The reference list is not as per journal requirements.
Author Response
Dear Prof.
We used reviewers and your insightful comments and we corrected parts of the article which were not meaningful (see following text) and the manuscript was entirely corrected according to the reviewer viewpoints. Please do not hesitate to ask us any questions about the submitted manuscript. This research, like many other scientific studies, has many weaknesses. I would like to express my deep gratitude to the hard-working referee of the Journal for his English language re-editing and wise-scientific re-judgment.
Sincerely yours,
Mohammad Goli
Hi dear editorial board and reviewer 3
Yours sincerely, thank you, the editor and reviewer 3, for your kind consideration for your scientific attention to my manuscript, for having read it, and for your valuable scientific and intelligent comments. I hope that by using your guidance, dear referees will attempt to refine the article and raise its scientific level. The yellow, green, and turquoise highlight in the uncleaned-revised manuscript related to the final reviewer 1, 2, and 3 proposed amendments, respectively.
Comments and Suggestions for Authors
Dear Authors,
The Current paper that you submitted to the Foods Journal it is interesting to the readers, and it is significant to the field, however, in this form, it's very messy, hard to follow, and is not prepared as per journal recommendations.
- Thank you very much for accepting the refereeing and scientific improvement of the article with your very wise and clever suggestions, dear referee. I tried to increase the scientific quality by making corrections suggested by you and two other dear referees for a better understanding of the article reader. All your suggestions, including how to refer in the text and in the reference list, were adjusted according to the guidelines of the journal.
The paper is extremely poor presented. Introduction chapter do not present sufficient relevant background information.
- Thank you so much for wise comment, therefore we amended introduction according to comments you and the other reviewers for presenting sufficient relevant background information as the following text:
- Introduction
Egg white contains a variety of proteins, including ovalbumin, ovotransferrin, ovomucoid, ovomucin, and lysozyme [1]. Due to its high nutritional content and good functional characteristics, egg white is an important component in the food industry. Egg white powder has become a popular commercial product as an alternative for fresh and liquid eggs because it can avoid microbiological and oxidative deterioration and has cheaper packaging costs [2]. Proteins' functional properties can be altered by physical, chemical, and enzymatic treatments. Since chemicals are hazardous to human health, the use of chemical techniques to modify proteins in the food industry is strictly prohibited [3]. The use of enzyme modification is often restricted due to the high cost of enzymes as well as difficult and complex operating conditions. As a result, physical protein modification (thermal processing, i.e., microwave treatment) is in considerable interest [4]. The current challenge is to identify novel processing techniques (such as ultrasound, pulsed electric field, high pressure processing, radio frequency, ultraviolet light, microwave, and cold plasma for egg products) that improve not only the intrinsic functional properties of eggs or their components, but also the product's quality. These innovative methods are recognized for their advantages over thermal treatments, particularly in protecting the heat sensitive nature of eggs and egg products. The availability of alternative processing methods has substantially increased the structural features, technological functionality, nutritional value, and safety of eggs and egg products [5].
Protein phosphorylation is a typical post-translational protein modification occurring in nature. In general, amino acid residues are phosphorylated by the addition of a covalent phosphate group bound by the protein kinase by serine, threonine, and tyrosine. Phosphorylation modifies the amino acid side chain by adding a charge and a hydrophilic group. According to various studies, phosphorylated proteins have higher solubility than non-phosphorylated proteins because they have a larger negative charge. The phosphorylated protein molecules have a strong electrical repulsion force, which increases their solubility in water. After egg white phosphorylation, powder particle size decreased but wettability, dispersibility, and solubility rose [2]. The most prevalent phosphorylation techniques are dry heating or bath water heating [6]. Microwave-assisted phosphorylation significantly accelerated the procedure, and phosphorylation of egg white protein resulted in improved functional properties. The effect of microwave-assisted phosphorylation on powder characteristics, on the other hand, is unclear [2].
For a few years, microwave-assisted chemical procedures have been employed to achieve chemical reactions. When compared to traditional heating techniques, chemical synthesis with microwave help can significantly shorten reaction duration while also improving yield and product purity. Microwave-assisted approaches have previously been used by certain researchers in the chemical alteration of dietary protein [7]. When compared to the traditional dry-heating approach, the microwave treatment can significantly shorten reaction times and speed the phosphorylation process. Li et al. [2] used a spray dryer to produce three different egg white protein powders (untreated, microwave-alone treated, and microwave-assisted phosphorylation modification by sodium tri-phosphate (STP or Na3PO4)) and investigated their physicochemistry and rehydration behavior.
There are several drying techniques available for producing sodium tri-phosphate egg white powder (STP-EWP), and these drying methods should be chosen based on drying efficiency and dried product quality. It has been found that freeze-drying proteins retain their natural structural shape better because they are subjected to fewer heat and water evaporation-related stresses. The freeze-dried particles are more porous and shrink less. According to Lili et al. [8] freeze-dried STP-EWP powders outperformed spray-dried STP-EWP powders in terms of solubility, emulsion stability, water holding capacity, oil and water absorption capacity, and heat gel strength. Freeze drying was found to be the most effective way for producing modified EWP powders with enhanced functional characteristics.
For the first time, the combined effects of phosphorylation with Na2HPO4 (1.5, 2.5, and 3.5 %), microwave treatment with power (200, 400, and 700 watt), and finally freeze drying (0.04-0.06 mbar & -50 ) on the functional characteristics including protein solubility, foam capacity, foam stability, emulsion activity, emulsion stability, water holding capacity , oil holding capacity, viscosity of solution 1%, as well as degree of phosphorylation, particle-size distribution, FTIR phosphorylation analysis, and SEM morphological analysis of egg white protein powder were investigated.
In the entire paper, use % the symbol instead of percent.
- Thank you for your very subtle and intelligent comments. All your suggestions (percentage words) in the text were converted to %. Please consider the “Revised-foods-1865422” file.
In the text, reference numbers should be placed in square brackets [ ], and placed before the punctuation; for example [1], [1–3] or [1,3]. For embedded citations in the text with pagination, use both parentheses and brackets to indicate the reference number and page numbers; for example [5] (p. 10). or [6] (pp. 101–105).
- All your suggestions, including how to refer in the text and in the reference list, were adjusted according to the guidelines of the journal. Please consider the “Revised-foods-1865422” file.
The results are not very clear presented, and all the information is presented in only two tables, and is very hard to understand it. I suggest to split the tables into smallest ones, which are much more clear and logical.
- Thank you for so much attention. It is interesting to you, very respected and scientific referee, that due to the limitations of the Figs and Tables and better understanding of the readers of the article, the authors of the article intended to divide all the information of the article into the following four categories.
- 1- Functional properties of egg white protein in Table 1
- 2- Particle-size distribution of egg white protein in Table 2
- 3- FTIR of egg white protein in Figure 1
- 4- SEM of egg white protein in Figure 2
- Please express your satisfaction with this editing and data classification of the article, because in our opinion, this category is more suitable for the authors. Please let us know if you do not agree with the opinions of our authors and you are not satisfied, so that I will inevitably correct it in the next revision
The authors mention in the paper Figure 2, but this figure is missing from the paper.
- I really enjoyed your thoughtfulness. Unfortunately, I don't know why the figure 2 was not visible to you, dear referee, while I had presented it in the submitted article for the journal. However, figure 2 was added to the text of the article. Please pay attention to file “Revised-foods-1865422”
The discussions are not sufficient.
- Thank you so much dear reviewer for improving our article. Please consider the following subtitles presented in revised manuscript specially yellow highlights in discussion section:
- 1.1. PM-EWP Protein solubility
- PM-EWP viscosity solution 1% (w/v)
- 2.1. PM-EWP particle size
- 2.2. PM-EWP Fourier transform infrared spectroscopy (FTIR)
Figure 1, is not understandable. Is poorly presented, please replace it with another one which is much more clear.
- The image quality of Figure 1 has improved in terms of clarity. Caption added for better understanding (Please consider the following :). In addition, these different subfigures were mentioned in the text of the article.
Fig 1. Fourier transform infrared spectroscopy (FTIR) of egg white powder treated with different phosphorylation (%) and microwave powers (W)
A, B, and C compare the effect of different microwave powers on 1.5, 2.5, and 3.5 % phosphorylation, respectively. D, E and F compare the effects of different percentages of phosphorylation in microwave powers of 200, 400 and 700 watts, respectively.
The reference list is not as per journal requirements.
- Thank you so much for your consideration of the Journal’s guidelines. Please pay attention to the “Revised-foods-1865422” file and the following text:
References
- Sheng, L.; Wang, Y.; Chen, J.; Zou, J.; Wang, Q.; Ma, M. Influence of highintensity ultrasound on foaming and structural properties of egg white. Food Res. Int. 2018, 108, 604-610.
- Li, P.; Jin, Y.; Sheng, L. Impact of microwave assisted phosphorylation on the physicochemistry and rehydration behaviour of egg white powder. Food Hydrocoll. 2020, 100, 105380.
- Mirmoghtadaie, L.; Shojaee-Aliabadi, S.; Hosseini, S.M. Recent approaches in physical modification of protein functionality. Food Chem. 2016, 199, 619-627.
- Ashokkumar, M.; Sunartio, D.; Kentish, S.; Mawson, R.; Simons, L.; Vilkhu, K.; Versteeg, C.K. Modification of food ingredients by ultrasound to improve functionality: A preliminary study on a model system. Food Sci. Emerg. Technol. 2008, 9, 155-160.
- Afraz, M.T.; Khan, M.R.; Roobab, U.; Noranizan, M.A.; Tiwari, B.K.; Rashid, M.T.; Inam-ur-Raheem, M.; Hashemi, S.M.B.; Aadil, R.M. Impact of novel processing techniques on the functional properties of egg products and derivatives: A review. Food Process Eng. 2020, 43, e13568.
- Pelegrine, D.H.G.; Gasparetto, C.A. Whey proteins solubility as function of temperature and pH. LWT - Food Sci. Technol. 2005, 38, 77-80.
- Li, P.; Sun, Z.; Ma, M.; Jin, Y.; Sheng, L. (2018). Effect of microwave-assisted phosphorylation modification on the structural and foaming properties of egg white powder. LWT - Food Sci. Technol. 2018, 97, 151-156.
- Lili, L.; Huan, W.; Guangyue, R.; Xu, D.; Dan, L.; Guangjun, Y. Effects of freeze-drying and spray drying processes on functional properties of phosphorylation of egg white protein. J. Agric. Biol. Eng. 2015, 8: 116-123.
- Patrignani, F.; Vannini, L.; Sado-Kamdem, S. L.; Hernando, I.; Marco-Moles, R.; Guerzoni, M. E.; Lanciotti, R. High pressure homogenization vs heat treatment: Safety and functional properties of liquid whole egg. Food Microbiol, 2013, 26, 63-69.
- Salvador, P.; Toldra, M.; Saguer, E.; Carretero, C.; Paresl, D. (2009). Microstructure-function relationships of heat-induced gels of porcine haemoglobin. Food Hydrocoll. 2009, 23, 1654-1659.
- Stefanovic, A.; Jovanovic, J.; Dojcinovic, M.; Levic, S.; Zuza, M.; Nedovic, V.; Knezevic-Jugovic, Z. Impact of high-intensity ultrasound probe on the functionality of egg white proteins. Hyg. Eng. Des. 2014, 6, 215-224.
- Neto, V-Q.; Narain, N.; Silvia, J-B.; Bora, P.S. Functional properties of raw and heat-processed cashew nut (Anarcardium occidentale L.) kernel protein isolate. Nutr. Food Res. 2001, 45, 258-262.
- Segura-Campos, M.; Perez-Hernandez, R.; Chel-Guerrero, L.; Castellanos-Ruelas, A.; Gallegos-Tintore, S.; Betancur-Ancona, D. (2013). Physicochemical and Functional Properties of Dehydrated Japanese Quail (Coturnix japonica) Egg White. Food Nutr. Sci. 2013, 4, 289-298.
- Li, P.; Sheng, L.; Jin, Y. (2019). Using microwave-assisted phosphorylation to improve foaming and solubility of egg white by response surface methodology. Poultry Sci. 2019, 98, 7110-7117.
- Afkhami, R.; Goli, M.; Keramat, J. Functional orange juice enriched with encapsulated polyphenolic extract of lime waste and hesperidin. J. Food Sci. 2018, 53, 634-643.
- Zaghian, N.; Goli, M. (2020). Optimization of the production conditions of primary (W1/O) and double (W1/O/W2) nano-emulsions containing vitamin B12 in skim milk using ultrasound wave by response surface methodology. Food Meas. Charact. 2020, 14, 3216-3226.
- Schmidmeier, C.; Wen, Y.; Drapala, K.P.; Dennehy, T.; McGuirke, A.; Cronin, K.; O'Mahony, J.A. (2019). The effect of agglomerate integrity and blending formulation on the mechanical properties of whey protein concentrate powder tablets. Food Eng. 2019, 247, 144-151.
- Maghamian, N.; Goli, M.; Najarian, A. (2021). Ultrasound-assisted preparation of double nano-emulsions loaded with glycyrrhizic acid in the internal aqueous phase and skim milk as the external aqueous phase. LWT - Food Sci. Technol. 2021, 141, 108-109.
- Jun, S.; Yaoyao, M.; Hui, J.; Obadi, M.; Zhongwei, C.; Bin, X. Effects of single- and dual-frequency ultrasound on the functionality of egg white protein. Food Eng. 2020, 277, 109902.
- Sheng, L.; Huang, M.; Wang, J.; Xu, Q.; Hammad, H.H.M.; Ma, M. (2018). A study of storage impaction ovalbumin structure of chicken egg. Food Eng. 2018, 212, 1-7.
- Pelegrine, D.H.G.; Gasparetto, C.A. Whey proteins solubility as function of temprature and pH. Food Sci. Technol. 2005, 38, 77-80.
- Stefanovic, A.B.; Jovanovic, J.R.; Grbavcic, S-Z.; Sekuljica, N. Z.; Manojlovic, V.B.; Bugarski, BM.; Knezevic-Jugovic, Z.D. Effect of the controlled high-intensity ultrasound on improving functionality and structural changes of egg white proteins. Food Bioproc Tech. 2017, 10, 1224-1239.
- Higuera-Barraza, O.A.; Toro-Sanchez, C.L.D.; Ruiz-Cruz, S.; Márquez-Rios, E. Effects of high-energy ultrasound on the functional properties of proteins. Ultrason Sonochem. 2016, 31, 558-562.
- Li, CP.; Ibrahim, HR.; Sugimoto, Y.; Hatta, H.; Aoki, T. Improvement of functional properties of egg white protein through phosphorylation by dryheating in the presence of pyrophosphate. Food Chem. 2004, 74, 5752-5758.
- Zhou, B.; Zhang, M.; Fang, Z.; Liu, Y. (2014). A combination of freeze drying and microwave vacuum drying of Duck egg white protein powders, Technol. 2014, 32, 1840-1847.
- Dengle Duan, D.; Ruan, R.; Wang, Y.; Liu, Y.; Dai, L.; Zhao, Y.; Zhou, Y.; Wu, Q. Microwave-assisted acid pretreatment of alkali lignin: Effect on characteristics and pyrolysis behavior. Technol. 2018, 251, 57-62.
- Dombrowski, J.; Johler, F.; Warncke, M.; Kulozik, U. Correlation between bulk characteristics of aggregated β-lactoglobulin and its surface and foaming properties. Food Hydrocoll. 2016, 61, 318-328.
- Soria, A.C.; Villamiel, M. Effect of ultrasound on the technological properties and bioactivity of food: a review. Trends Food Sci. Technol. 2010, 21, 323-331.
- Stanciuc, N.; Banu, I.; Turturica, M.; Aprodu, I. pH and heat induced structural changes of chicken ovalbumin in relation with antigenic properties. J. Biol. Macromol. 2016, 93, 572-581.
- Wang, X.; Gu, L.; Su, Y.; Li, J.; Yang, Y.; Chang, C. Microwave technology as a new strategy to induce structural transition and foaming properties improvement of egg white powder. Food Hydrocoll. 2020, 101, e105530.
- Wang, X.; Chi, Y-J. Microwave-assisted phosphorylation of soybean protein isolates and their physicochemical properties. Czech J. Food Sci. 2012, 30, 99-107.
- Hayashi, Y.; Li, C.P.; Enomoto, H.; Ibrahim, H.R.; Sugimoto, Y.; Aoki, T. Improvement of functional properties of ovotransferrin by phosphorylation through dry-heating in the presence of pyrophosphate. Asian-Australas J Anim Sci. 2008, 21, 596-602.
- Xiong, Z.; M, Ma. Enhanced ovalbumin stability at oilwater interface by phosphorylation and identification of phosphorylation site using maldi-tof mass spectrometry. Colloids Surf. B. 2017, 153, 253-262.
- Arzeni, C.; Perez, O. E.; Pilosof, A.M.R. Functionality of egg white proteins as affected by high intensity ultrasound. Food Hydrocoll. 2012, 29, 308-316.
- Hadidi, M.; Jafarzadeh, S.; Ibarz, A. Modified mung bean protein: Optimization of microwave-assisted phosphorylation and its functional and structural characterizations. LWT-Food Sci. Technol. 2021, 151, 1-2.
- Tang, S.; Yu, J.; Lu, L.; Fu, X.; Cai, Z. Interfacial and enhanced emulsifying behavior of phosphorylated ovalbumin. J. Biol. Macromol. 2019, 131, 293-300.
- Sheng, L.; Su, P.; Han, K.; Chen, J.; Cao, A.; Zhang, Z.; Jin, y.; Ma, M. Synthesis and structural characterization of lysozyme–pullulan conjugates obtained by the maillard reaction. Food Hydrocoll. 2017, 71, 1-7.

Round 2
Reviewer 1 Report
Author has revised and responded according to the given comments.
However, regarding the comments about changing from table form of results to figure form, has not been addressed, author mentioned about the page limitation, as Foods journal has not limit to page count, and it is open access, online, so there wont be any trouble.
So please change results presentation from table into figures, as it will help a lot for understanding the results with trend clearly.
Author Response
Dear Prof.
We used reviewers and your insightful comments and we corrected parts of the article which were not meaningful (see following text) and the manuscript was entirely corrected according to the reviewer viewpoints. Please do not hesitate to ask us any questions about the submitted manuscript. This research, like many other scientific studies, has many weaknesses. I would like to express my deep gratitude to the hard-working referee of the Journal for his English language re-editing and wise-scientific re-judgment.
Sincerely yours,
Mohammad Goli
Hi dear editorial board and reviewer 1
Yours sincerely, thank you, the editor and reviewer 1, for your kind consideration for your scientific attention to my manuscript, for having read it, and for your valuable scientific and intelligent comments. I hope that by using your guidance, dear referees will attempt to refine the article and raise its scientific level. The yellow, and turquoise highlight in the uncleaned-revised manuscript related to the final reviewer 1, and 3 proposed amendments, respectively.
Comments and Suggestions for Authors
Author has revised and responded according to the given comments.
- Thank you so much dear reviewer1. Undoubtedly, your very scientific comments have been effective in improving the article
However, regarding the comments about changing from table form of results to figure form, has not been addressed, author mentioned about the page limitation, as Foods journal has not limit to page count, and it is open access, online, so there wont be any trouble.
So please change results presentation from table into figures, as it will help a lot for understanding the results with trend clearly.
- I converted Tables 1 and 2 into a relevant figures according to your previous comment.

Reviewer 3 Report
Dear Authors,
I understand that English is not your first language; however, I am not judging from this point of view, I am judging the content of the paper. Please find some more specific comments below and some suggestion that I would like to recommend. Although is clear that the authors made some great efforts, there are still some changes that need to be made.
Obs 1. In the abstract, the second sentence is too long. Please find my suggestion. Also, try to use the past tense instead of present simple when explain your results.
P1.5-M200 had the highest oil holding capacity, emulsion stability, and emulsion activity, while P2.5-M200 had the highest foam capacity. The P2.5-M400 had the largest particle size, and P3.5-M200 had the highest degree of phosphorylation and protein solubility. On the other hand, P3.5-M200 has the highest solution viscosity by 1% (w/v), water holding capacity, and foam stability, in the treatments that used phosphorylation and microwave simultaneously.
Obs 2. Introduction, second sentence (row 38-40), I suggest the following correction
Eggs are highly nutritious presenting many functional properties for human consumption (use a reference here); however, egg white is the most important component for the food industry (Li et al. 2020).
maybe this one is good https://doi.org/10.1038/s41598-021-00343-1
Li, P.; Jin, Y.; Sheng, L. Impact of microwave assisted phosphorylation on the physicochemistry and rehydration behav-iour of egg white powder. Food Hydrocoll. 2020, 100, 105380.
Obs 3. Avoid repetitions of same words in the same sentence. For example on row 45 ( The techniques are not chemicals … )
Since chemicals are hazardous to human health, their usage to modify proteins in the food industry is strictly prohibited.
Obs 4. Row 49 … is of considerable …
Obs 5. Row 65. According to various studies … what studies? Please use some references . Here you can find some.
Sun, N., Wang, Y., Bao, Z., Cui, P., Wang, S., & Lin, S. (2020). Calcium binding to herring egg phosphopeptides: Binding characteristics, conformational structure and intermolecular forces. Food chemistry, 310, 125867.
Bilbrough, T., Piemontese, E., & Seitz, O. (2022). Dissecting the role of protein phosphorylation: a chemical biology toolbox. Chem. Soc. Rev, 51, 5691-5730.
Obs 6. Material and methods, row 112. How many eggs were analysed? Please add the number of samples.
Row 118. The white of the fresh eggs was carefully separated from the yolk.
Row 193 – 200, please use the same writing font, here is different.
Row 277 – 288. In this part the authors explain the effect of protein solubility and hydrophobicity, but in table 1 the protein hydrophobicity is not given, only the protein solubility.
Row 269 to 309. The authors talk about foaming ability and stability, but in table 1 is given the foaming capacity and stability, as well as in subchapter 3.1.2. PM-EWP foaming ability and stability. Please correct this.
Row 345- 346. …. SEM observations and particle physical characteristics considerable in Table 2. I really don’t understand this part. What do you mean in table 2?
Row 347. In table 2. Please use dot instead of comma d(3.2) and d(4.3) (μm)
Row 361-365. The following sentence I suggest this modifications:
This disparity might be explained by the following two factors: First, in the control group, protein molecules with a tight and regular structure were largely dispersed in the same flow layer of solution, and there was no evident entanglement between them. Second, the flexible protein molecules were placed in distinct flow levels of solution and entangled after microwave treatment. This effect was also previously observed by other authors (Wang et al. 2020).
Wang, X.; Gu, L.; Su, Y.; Li, J.; Yang, Y.; Chang, C. Microwave technology as a new strategy to induce structural transition and foaming properties improvement of egg white powder. Food Hydrocoll. 2020, 101, e105530.
Row 412. index d(3.2) point instead of comma
Row 426. index d(4.3) same
Row 431 , please do not use i.e when mentioning you data and your results, instead you can rephrase it as following:
In the current study, higher microwave power, (700 watts), resulted in a significant reduction in the protein particle size of many animal and vegetable proteins tested. This result can be explained by the breakdown of non-covalent associative forces such as hydrogen bonding, hydrophobic, and electrostatic interactions that keep lower protein aggregation in solution. Recently, other authors reported similar effect when the effect of microwave assisted phosphorylation on the physicochemistry and rehydration behaviour of egg white powder was tested [2, 7 Li et al. 2018, 2020].
Li, P.; Sun, Z.; Ma, M.; Jin, Y.; Sheng, L. (2018). Effect of microwave-assisted phosphorylation modification on the struc-tural and foaming properties of egg white powder. LWT - Food Sci. Technol. 2018, 97, 151-156.
Li, P.; Jin, Y.; Sheng, L. Impact of microwave assisted phosphorylation on the physicochemistry and rehydration behav-iour of egg white powder. Food Hydrocoll. 2020, 100, 105380.
I am glad to see figure 2 added.
General comment. Please be careful that references 2 and 7 are too many times cited through the paper. Maybe the authors can find others also.
Author Response
Dear Prof.
We used reviewers and your insightful comments and we corrected parts of the article which were not meaningful (see following text) and the manuscript was entirely corrected according to the reviewer viewpoints. Please do not hesitate to ask us any questions about the submitted manuscript. This research, like many other scientific studies, has many weaknesses. I would like to express my deep gratitude to the hard-working referee of the Journal for his English language re-editing and wise-scientific re-judgment.
Sincerely yours,
Mohammad Goli
Hi dear editorial board and reviewer 3
Yours sincerely, thank you, the editor and reviewer 3, for your kind consideration for your scientific attention to my manuscript, for having read it, and for your valuable scientific and intelligent comments. I hope that by using your guidance, dear referees will attempt to refine the article and raise its scientific level. The yellow, and turquoise highlight in the uncleaned-revised manuscript related to the final reviewer 1, and 3 proposed amendments, respectively.
Comments and Suggestions for Authors
Dear Authors,
I understand that English is not your first language; however, I am not judging from this point of view, I am judging the content of the paper. Please find some more specific comments below and some suggestion that I would like to recommend. Although is clear that the authors made some great efforts, there are still some changes that need to be made.
- Thank you for your understanding and kindness in correcting the article as much as possible. Thank you very much for all your efforts dear referee
Obs 1. In the abstract, the second sentence is too long. Please find my suggestion. Also, try to use the past tense instead of present simple when explain your results.
P1.5-M200 had the highest oil holding capacity, emulsion stability, and emulsion activity, while P2.5-M200 had the highest foam capacity. The P2.5-M400 had the largest particle size, and P3.5-M200 had the highest degree of phosphorylation and protein solubility. On the other hand, P3.5-M200 has the highest solution viscosity by 1% (w/v), water holding capacity, and foam stability, in the treatments that used phosphorylation and microwave simultaneously.
- Your sentences and suggestions were implemented exactly and replaced with the previous text.
Obs 2. Introduction, second sentence (row 38-40), I suggest the following correction
Eggs are highly nutritious presenting many functional properties for human consumption (use a reference here); however, egg white is the most important component for the food industry (Li et al. 2020).
maybe this one is good https://doi.org/10.1038/s41598-021-00343-1
Li, P.; Jin, Y.; Sheng, L. Impact of microwave assisted phosphorylation on the physicochemistry and rehydration behav-iour of egg white powder. Food Hydrocoll. 2020, 100, 105380.
- Your comment was done according to the follow:
Eggs are highly nutritious presenting many functional properties for human consumption [2]; however, egg white is the most important component for the food industry [3].
Obs 3. Avoid repetitions of same words in the same sentence. For example on row 45 ( The techniques are not chemicals … )
Since chemicals are hazardous to human health, their usage to modify proteins in the food industry is strictly prohibited.
- Thank you so much for the wise comment and it was amended according to your suggestion. Please pay attention to the “Revised-2 file” attached.
Obs 4. Row 49 … is of considerable …
Obs 5. Row 65. According to various studies … what studies? Please use some references . Here you can find some.
Sun, N., Wang, Y., Bao, Z., Cui, P., Wang, S., & Lin, S. (2020). Calcium binding to herring egg phosphopeptides: Binding characteristics, conformational structure and intermolecular forces. Food chemistry, 310, 125867.
Bilbrough, T., Piemontese, E., & Seitz, O. (2022). Dissecting the role of protein phosphorylation: a chemical biology toolbox. Chem. Soc. Rev, 51, 5691-5730.
- These two references were consider in manuscript text according to your comment as the following:
According to various studies, phosphorylated proteins have higher solubility than non-phosphorylated proteins because they have a larger negative charge [7, 8].
Obs 6. Material and methods, row 112. How many eggs were analysed? Please add the number of samples.
- It was done according your wise comment as the follow:
- 360 fresh chicken eggs were provided from a local farm (Simorgh LTD, Isfahan, Iran).
Row 118. The white of the fresh eggs was carefully separated from the yolk.
- It was done thank you so much for improving the manuscript text.
Row 193 – 200, please use the same writing font, here is different.
- Thank you so much. It was done as your comment.
Row 277 – 288. In this part the authors explain the effect of protein solubility and hydrophobicity, but in table 1 the protein hydrophobicity is not given, only the protein solubility.
- Thank you so much for wise and scientific comment and it was done according to the follow:
Protein solubility is the important factor in protein foaming [23]. Partially opening the protein structure during the microwave process, exposing the hydrophobic and sulfhydryl groups, or partially denaturing the protein, reducing particle size and allows for better protein absorption at the air-water interface, increasing the protein's foaming properties [23, 26]. Proteins with a high surface hydrophobicity were identified to adsorb extensively at the air-water interface, resulting in a considerable reduction in interface or surface tension, which was used for foaming [17]. The electrical conductivity of microwave-vacuum drying EWP samples may have also been increased; as a result, the distribution of charged, polar, and non-polar residues of the protein molecules was altered, reducing the foaming capacity of the microwave-vacuum drying protein compared to freeze drying samples [27, 28].
Row 269 to 309. The authors talk about foaming ability and stability, but in table 1 is given the foaming capacity and stability, as well as in subchapter 3.1.2. PM-EWP foaming ability and stability. Please correct this.
- Thank you so much for wise and scientific comment and it was done according to the follow:
3.1.2. PM-EWP foaming capacity and stability
Fig. 2 shows that increasing the microwave power from 200 to 700 watts decreased the foam production capacity significantly at each level of phosphorylation, and the treatment containing 2.5 % phosphorylation had the highest foam production capacity at each level of microwave power. Treatments P2.5-M200 and P0-M700 had the highest and lowest foam production capacity, respectively. The dispersion of gas bubbles inside a continuous liquid or semi-solid phase results in foam in food [25]. Protein solubility is the important factor in protein foaming [23]. Partially opening the protein structure during the microwave process, exposing the hydrophobic and sulfhydryl groups, or partially denaturing the protein, reducing particle size and allows for better protein absorption at the air-water interface, increasing the protein's foaming properties [23, 26]. Proteins with a high surface hydrophobicity were identified to adsorb extensively at the air-water interface, resulting in a considerable reduction in interface or surface tension, which was used for foaming [17]. The electrical conductivity of microwave-vacuum drying EWP samples may have also been increased; as a result, the distribution of charged, polar, and non-polar residues of the protein molecules was altered, reducing the foaming capacity of the microwave-vacuum drying protein compared to freeze drying samples [27, 28].
Increasing the microwave power from 400 to 700 watts greatly enhanced the stability of the foam at each level of phosphorylation %, although the difference between different levels of microwave power of 200 and 400 watts was not significant. At all the same levels of applied microwave power, 3.5% and 2.5% phosphorylation showed the highest and lowest foam stability, respectively. P0-M700 and P3.5-M0 treatments had the highest and lowest foam stability, respectively. According to the statistics, the group with the greatest foaming capacity did not have the best foaming stability. Excessive microwave power might be to responsible for the findings. According to Duan et al. [29], extensive oxidation of egg white protein can result in an increase in foaming capacity but a decrease in foaming stability. The reduction in foaming stability could be due to a drop in the substituents stabilized and more cohesive interfacial coatings formed in the presence of bigger diameter aggregates, which led to a decrease in coalescence rate and hence foam degradation [30].
Row 345- 346. …. SEM observations and particle physical characteristics considerable in Table 2. I really don’t understand this part. What do you mean in table 2?
- This wise comment was done and we amended it according the following:
The water holding capacity and wetting behavior were affected by particle size and surface tension, with the smaller particles being less likely to be wetted. The PM-EWP samples exhibited a more porous structure, but the lowest particle size and highest specific surface area [3].
Row 347. In table 2. Please use dot instead of comma d(3.2) and d(4.3) (μm)
- It was completed in accordance with the scientific comment. Please keep in mind that Revewer-1 has insistently asked me to change two tables to relevant figures.
Row 361-365. The following sentence I suggest this modifications:
This disparity might be explained by the following two factors: First, in the control group, protein molecules with a tight and regular structure were largely dispersed in the same flow layer of solution, and there was no evident entanglement between them. Second, the flexible protein molecules were placed in distinct flow levels of solution and entangled after microwave treatment. This effect was also previously observed by other authors (Wang et al. 2020).
Wang, X.; Gu, L.; Su, Y.; Li, J.; Yang, Y.; Chang, C. Microwave technology as a new strategy to induce structural transition and foaming properties improvement of egg white powder. Food Hydrocoll. 2020, 101, e105530.
- Thank you so much for the scientific and intelligent comment. It was done according your suggestion as the follow:
The viscosity of PM-EWP solution 1% (w/v) under various conditions is shown in Fig. 5. The effect of increasing the microwave power to 700 watts on viscosity was not significant (P> 0.05). At varying shear rates, the viscosity of egg white power solution in the control group was essentially the same, demonstrating Newtonian fluid properties. While all egg white powder solutions following microwave treatment exhibited shear thinning behavior, indicating non-newtonian fluid properties. This disparity might be explained by the following two factors: First, in the control group, protein molecules with a tight and regular structure were largely dispersed in the same flow layer of solution, and there was no evident entanglement between them. Second, the flexible protein molecules were placed in distinct flow levels of solution and entangled after microwave treatment. This effect was also previously observed by other authors [33].
Row 412. index d(3.2) point instead of comma
- It was done.
Row 416. index d(4.3) same
- It was done.
Row 431 , please do not use i.e when mentioning you data and your results, instead you can rephrase it as following:
In the current study, higher microwave power, (700 watts), resulted in a significant reduction in the protein particle size of many animal and vegetable proteins tested. This result can be explained by the breakdown of non-covalent associative forces such as hydrogen bonding, hydrophobic, and electrostatic interactions that keep lower protein aggregation in solution. Recently, other authors reported similar effect when the effect of microwave assisted phosphorylation on the physicochemistry and rehydration behaviour of egg white powder was tested [2, 7 Li et al. 2018, 2020].
Li, P.; Sun, Z.; Ma, M.; Jin, Y.; Sheng, L. (2018). Effect of microwave-assisted phosphorylation modification on the struc-tural and foaming properties of egg white powder. LWT - Food Sci. Technol. 2018, 97, 151-156.
Li, P.; Jin, Y.; Sheng, L. Impact of microwave assisted phosphorylation on the physicochemistry and rehydration behav-iour of egg white powder. Food Hydrocoll. 2020, 100, 105380.
- It was done and the two references i.e., 2 and 7 were changed to 3 and 10, respectively.
In the current study, higher microwave power, (700 watts), resulted in a significant reduction in the protein particle size of many animal and vegetable proteins tested. This result can be explained by the breakdown of non-covalent associative forces such as hydrogen bonding, hydrophobic, and electrostatic interactions that keep lower protein aggregation in solution. Recently, other authors reported similar effect when the effect of microwave assisted phosphorylation on the physicochemistry and rehydration behaviour of egg white powder was tested [3, 10].
I am glad to see figure 2 added.
- It was done. Of course it was added to the “revised-1 file” but I don’t know why not you saw. I considered it with the Fig. 9 in the “revised-2 file” resubmitted.
General comment. Please be careful that references 2 and 7 are too many times cited through the paper. Maybe the authors can find others also.
- Thank you for your attention, but really, the research on the effect of microwaves and phosphorylation on egg white has been done mainly by Lee and his colleagues in recent years. Of course, I included the three references suggested by you, dear referee, in this article. I hope you are satisfied with it.
- Dear Reviewer 3, please be aware that, at the request of Reviewer 1, I forcefully transformed two existing tables into seven figures, resulting in a total of figures in the article to nine. Please pardon me because the figures were a little scrambled when they were presented in the Foods style. Certainly, these figures will be optimally modified and corrected during the publication's final proof.
